# LLM4Branch: Large Language Model for Discovering Efficient Branching Policies of Integer Programs

Zhinan Hou [* 1]  Xingchen Li [* 1]  Yankai Zhang [1]  Tianxun Li [1]  Keyou You [1]

## Abstract

Efficient branching policies are essential for accelerating Mixed Integer Linear Programming (MILP) solvers. Their design has long relied on hand-crafted heuristics, and now machine learning has emerged as a promising paradigm to automate this process. However, existing learning-based methods are often hindered by their dependence on expensive expert demonstrations and the gap between training objectives and the solver's end-to-end performance. In this work, we propose LLM4Branch, a novel framework that leverages Large Language Models (LLMs) to automate the discovery of efficient branching policies. Specifically, the discovered policy is an executable program with a program skeleton generated by the LLM and a parameter vector, which is optimized via a zeroth-order method over a few instances with their end-to-end performance feedback. Extensive experiments on standard MILP benchmarks demonstrate that LLM4Branch establishes a new state-of-the-art among CPU-based methods and achieves performance competitive with advanced GPU-based models. Codes are available at https://github.com/hzn18/LLM4Branch.

## 1. Introduction

Mixed integer linear programming (MILP) is fundamental for decision-making in diverse domains such as logistics (Alnahhal et al., 2021), vehicle routing (Yao et al., 2021), and energy management (Dukovska et al., 2021). Many important MILP problems are NP-hard, and solving large-scale instances to optimality remains a computational challenge. As problem sizes grow, the exponential expansion of the search space makes finding optimal solutions within practi-

cal time limits a significant difficulty.

The Branch-and-Bound (B&B) algorithm (Lodi & Zarpellon, 2017) is the cornerstone of modern exact MILP solvers. It explores the solution space by recursively *branching* on integer variables to build a tree of subproblems, while using *bounding* techniques to prune branches, many of which do not contain the optimal solution. The branching policy, which determines the variable to branch on at each node, is critical to its performance. An effective policy can dramatically reduce the search tree size and accelerate convergence, whereas a poor one can lead to an exponential increase in computation time.

Research on branching policies has followed two main paradigms. The first involves hand-crafted heuristics, such as strong branching (Applegate et al., 1995; Dey et al., 2024) and pseudocost branching (Linderoth & Savelsbergh, 1999). Although these policies demonstrate effectiveness across a wide range of problem types, designing such heuristic methods requires substantial domain-specific expertise. Moreover, they are typically designed for general-purpose usage and consequently fail to exploit distribution-specific patterns that could yield higher performance on specialized problem instances (Bengio et al., 2021).

The second paradigm leverages machine learning to construct data-driven branching policies, where Imitation Learning (IL) constitutes the dominant approach (Gasse et al., 2019; Gupta et al., 2020; Kuang et al., 2024). Despite the rapid integration of these methods into solver design, they still face a critical barrier to practical deployment as they lack end-to-end optimization. Even state-of-the-art imitation models inherently suffer from *objective mismatch*, a phenomenon where the model optimizes a surrogate objective (e.g. the accuracy of imitating strong branching (Kuang et al., 2024)) rather than solver's end-to-end performance. Consequently, achieving high imitation accuracy does not reliably translate to enhanced solving efficiency (Gasse et al., 2022b). In this work, we are interested in *bypassing these surrogate limitations and discovering efficient branching policies,* which becomes increasingly urgent for solving large-scale MILP instances.

To realize this vision, Reinforcement Learning (RL) at-

---

[*]Equal contribution  [1]Department of Automation, BNRist, Tsinghua University, Beijing, China.. Correspondence to: Keyou You <youky@tsinghua.edu.cn>.

*Proceedings of the $43^{rd}$ International Conference on Machine Learning*, Seoul, South Korea. PMLR 306, 2026. Copyright 2026 by the author(s).

tempts to directly leverage solver feedback such as the number of branching nodes or solving time to guide policy updates. While this theoretically aligns the learning objective with the solver performance metric, utilizing such feedback in B&B presents severe practical hurdles (Parsonson et al., 2023). The sparse and delayed nature of rewards, where the effect of a decision is only revealed after thousands of steps, creates a difficult temporal credit assignment problem. This typically leads to extreme sample inefficiency and training instability.

Recent advances in large language models (LLMs) offer a transformative solution to this feedback utilization challenge (Novikov et al., 2025; Romera-Paredes et al., 2024; Shojaee et al., 2025). Unlike RL approaches that require extensive interaction to learn, modern LLMs can adapt to new problem instances through interactive reasoning without relying on exhaustive trial-and-error. This capability makes it promising to synthesize high-level reasoning with end-to-end solver feedback, allowing for the iterative evolution of efficient branching policies. Building on these insights, we propose LLM4Branch, a framework that leverages LLMs to automate the discovery of branching policies by integrating end-to-end solver performance feedback. Specifically, we introduce a novel branching policy representation where the policy is formulated as an executable program. Exploiting this representation, we decompose the discovery process into two phases: an LLM generates the program skeleton to define the branching policy structure, while a zeroth-order method optimizes the parameter vector using direct solver performance feedback. This design yields policies that exhibit a compact, human-readable structure derived from the LLM, while achieving high performance through rigorously optimized parameter configurations. Notably, LLM4Branch operates without an expert teacher and has the potential to discover efficient branching policies beyond human intuition.

We validate the effectiveness of LLM4Branch on four standard and two challenging MILP benchmarks. The results demonstrate that LLM4Branch establishes a new state-of-the-art among purely CPU-based methods and achieves performance competitive with advanced GPU-based models, highlighting its potential for practical, hardware-efficient deployment.

## 2. Related Work

### 2.1. Learning to Branch

Significant research has focused on leveraging machine learning to approximate the time-consuming expert branching policy, hoping to accelerate the branching module within B&B solvers. For instance, a support vector machine variant was trained by using data collected from a strong branch-

ing expert (Khalil et al., 2016), while tree models were utilized to predict strong branching scores (Alvarez et al., 2017). As a representative advancement, a Graph Neural Network (GNN) model was proposed (Gasse et al., 2019) to approximate the strong branching policy by representing each branching node state as a bipartite graph, achieving significant benefits over the SCIP solver (Achterberg, 2009). Following this, various extensions have been developed to improve scalability (Nair et al., 2020) and augment training samples (Zhang et al., 2024). However, the success of these methods is highly dependent on the expert demonstrations.

To overcome the limitations of imitation learning, recent research has explored RL to learn branching policies from solver feedback. For instance, prior works have employed deep Q-networks to approximate subtree sizes (Etheve et al., 2020) or decomposed search trees into shorter learning paths (Parsonson et al., 2023). More recently, SORREL (Feng & Yang, 2025) combined offline pre-training on heuristic demonstrations with self-imitation learning on past good experiences sampled by itself. Despite these innovative methods, applying RL to branching still necessitates extensive trial-and-error exploration.

While the existing works mentioned above demonstrate impressive performance, they often rely on GPU acceleration, which presents a significant barrier to practical deployment on the CPUs commonly used for modern exact MILP solvers. This has spurred research into the lightweight deployment of deep models. For example, a hybrid policy was proposed to mitigate computational costs by applying an expressive GNN only at the root node, while using a computationally efficient multi-layer perceptron for all subsequent nodes (Gupta et al., 2020). Alternatively, Symb4CO (Kuang et al., 2024) employs a deep sequential model to discover inherently lightweight symbolic branching policies by searching the space of mathematical expressions for effective rules.

### 2.2. Large Language Models for Integer Programs

The application of LLMs to integer programs has recently progressed along two primary research streams. The first stream leverages LLMs as creative engines within evolutionary frameworks to discover novel heuristic algorithms for specific optimization problems. FunSearch (Romera-Paredes et al., 2024) and AlphaEvolve (Novikov et al., 2025) pair LLMs with evaluators to iteratively refine programs for tasks like bin packing and job scheduling. To further enhance the search efficiency, recent studies have introduced advanced mechanisms: ReEvo (Ye et al., 2024) incorporates self-reflection to provide a verbal gradient for guidance, while HSEvo (Dat et al., 2025) introduces a diversity-driven harmony search to better balance exploration and exploitation. Furthermore, MCTS-AHD (Zheng et al., 2025) re-

structures the evolutionary process using Monte Carlo Tree Search to avoid local optima inherent in population-based methods. While these methods demonstrate remarkable success in discovering heuristics, their fundamental objective is to find high-quality solutions for specific problem domains.

In contrast, a more recent research direction employs LLMs to automate the design of internal components for integer programming solvers, aiming to accelerate the search for provably optimal solutions. For example, LLM-LNS (Ye et al., 2025) utilizes a dual-layer evolutionary agent to automate the crucial step of neighborhood selection within large neighborhood search. Similarly, EvoCut (Yazdani et al., 2025) addresses the complex process of cut generation by employing LLMs to initialize and refine candidate inequalities. Furthermore, LLM4Solver (Zhou et al., 2024) and DHEvo (Zhang et al., 2025) extend this automation to the design of diving heuristics, leveraging LLMs to synthesize executable heuristic algorithms. Collectively, these studies highlight a broadening of the field, demonstrating that LLMs can serve not only as designers of standalone heuristics but also as architects of core components for exact solvers.

## 3. Preliminaries

We consider the following MILP problem:

$$\min_{x \in \mathbb{R}^n} c^\top x \quad \text{s.t.} \quad Ax \le b, \ x \in \mathbb{Z}^p \times \mathbb{R}^{n-p},$$

where $c \in \mathbb{R}^n$ is the cost vector, $A \in \mathbb{R}^{m \times n}$ is the constraint matrix, $b \in \mathbb{R}^m$ is the right-hand side vector of the linear constraints, and $p$ is the number of integer variables.

The B&B algorithm serves as a fundamental framework for solving MILPs (Lodi & Zarpellon, 2017). It constructs a search tree whose the root represents the original problem and the other nodes correspond to subproblems obtained by tightening bounds on the variables. At each node, it solves the linear programming (LP) relaxation of the subproblem, obtained by disregarding the integrality constraints. If the resulting LP solution $x^*$ satisfies the integrality constraints, or is worse than the current best known integer solution, the node requires no further processing and is pruned. Otherwise, the problem is decomposed using the *branching candidate set* $\mathcal{C} = \{i \in \{1, \ldots, p\} \mid x_i^* \notin \mathbb{Z}\}$, which comprises all integer variables taking fractional values. From this set, it selects a specific *branching variable* $x_i \in \mathcal{C}$ to partitions the current node into two child nodes by imposing the constraints $x_i \le \lfloor x_i^* \rfloor$ and $x_i \ge \lceil x_i^* \rceil$, respectively. The algorithm proceeds by selecting an unexplored node and repeating these steps until the search is complete.

The selection of a branching variable from the candidate set $\mathcal{C}$ is a core component of the B&B algorithm, formally termed the *branching policy*. It significantly influences the total size of the search tree and, consequently, the overall

solving time. The goal of this work is to leverage LLMs to automate the discovery of efficient branching policy.

## 4. Method

The branching policy design aims to find a branching policy $p$ that minimizes the solver cost in a given set of MILP instances $\mathcal{D}$. Formally, we seek to minimize the following end-to-end performance metric:

$$M(p; \mathcal{D}) \coloneqq \frac{1}{|\mathcal{D}|} \sum_{d \in \mathcal{D}} m(p, d),$$

where $m(p, d)$ represents the solver cost (e.g., solving time, number of branching nodes or primal-dual gap) when applying branching policy $p$ to the instance $d$.

As mentioned before, learning-based approaches leverage strong branching as an expert oracle and aim to maximize the accuracy of imitating its decisions, which may not align with the end-to-end solver's performance metric $M(p; \mathcal{D})$, as empirically demonstrated in Appendix A. Moreover, they often employ neural networks as the branching policy $p$. They generally require GPU acceleration for efficient inference and act as black boxes, which hinders interpretability and practical integration. In this section, we propose LLM4Branch, a framework where we represent the branching policy as a human-readable, executable program and iteratively optimize it by directly minimizing the solver cost.

### 4.1. Overview of LLM4Branch

We represent the branching policy $p$ as an executable program, defined by a pair $p = (s, \boldsymbol{\theta})$, where $s$ is the program skeleton and $\boldsymbol{\theta}$ represents its parameter vector. This program maps a branching node representation to score vector, where each element represents the score for a corresponding candidate variable. The candidate associated with the maximum score is then selected for branching.

Following (Gasse et al., 2019; Kuang et al., 2024), we adopt a compact 91-dimensional feature vector as node representation. We selected this representation for its high computational efficiency, which enables purely CPU-based execution while maintaining comprehensive coverage. A detailed description of these features can be found in Appendix B.

The program skeleton $s$ is implemented as a parameterized function. It explicitly takes both features of candidate variable and the parameter vector as inputs to compute scores for candidate branching variables, structured as follows:

```
1  def score_function(variable_features, params):
2      """
3      Compute scores for each variables.
4
5      Input:
6      variable_features: (n_vars, n_features)
7      params: coefficient vector (params,)
```

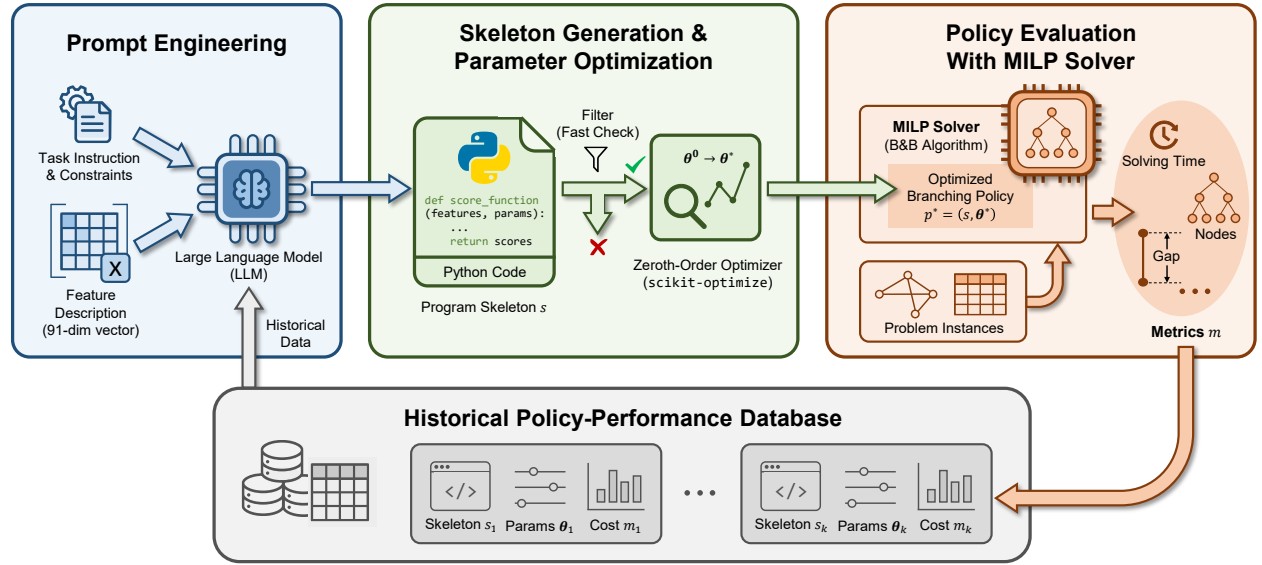

*Figure 1.* The overview of LLM4Branch.

```
 8
 9        Return: scores (n_vars,)
10        """
11
12        value_0 = variable_features[:, 0]
13        scores = params[0] * value_0 + params[1]
14        return scores
```

Deploying such a branching program offers distinct advantages over neural network models. First, the policy usually selects only a small subset of features, which significantly reduces the time spent on feature extraction, as we can skip the computation of unused features. Second, the policy involves very few parameters (often less than 10) and simple arithmetic operations. This makes it extremely lightweight in memory and much easier to optimize. A example of a generated branching program is provided in Appendix H.1.

We leverage reasoning and code generation capabilities of LLM to iteratively explore the program space of branching policy $p$. In each iteration, our approach samples a program skeleton $s$ from the LLM based on a structured prompt and feedback from previously generated branching programs. Then, the generated skeleton $s$ undergoes a rapid check, and proceed to the next stage only if satisfying a performance threshold. After that, we employ a zeroth-order method to optimize the parameter vector $\theta$, yielding an optimal parameter vector $\theta^*$ that minimize the solver cost on a subset $\mathcal{D}_{\text{sub}} \subseteq \mathcal{D}$. The resulting policy $p^* = (s, \theta^*)$ is then evaluated on the instance set $\mathcal{D}$ to obtain the final solver cost $M(p^*; \mathcal{D})$. These performance metrics are stored in the database to improve the next generation. The overview of LLM4Branch is illustrated in Figure 1. For the framework's detailed pseudocode, see Appendix D.2.

### 4.2. Skeleton Generation of the Branching Policy

The space of branching programs is vast. To explore this space efficiently, our method leverages LLMs as a core engine for skeleton generation. This generation process is organized as two key components: a structured prompt that guides the LLM using the feature inputs, and an evolutionary loop that uses historical policy-performance to improve the skeleton generation.

We leverage the in-context learning capabilities of LLMs to generate executable program skeletons. The prompt provided to the LLM is carefully structured with four key components: (a) **Task Instruction:** A high-level instruction to design a Python function that scores branching candidate variables based on 91-dimensional input features and a parameter vector. (b) **Feature Description:** A structured description of the 91-dimensional features, explaining their semantic meaning. (c) **Historical Feedback:** A context set of previously generated programs alongside their evaluated solver costs $M(p, d)$. (d) **Constraints and Format:** Explicit constraints are given, such as a limit on the total number of features used (e.g., at most 10) to promote simplicity, and a requirement to output syntactically correct Python code. The full prompt is provided in Appendix D.1.

Our method operates as an evolutionary loop guided by performance feedback. The process simply begins with a program that selects a branching variable at random from the branching candidate set. In each iteration, the program undergoes parameter optimization (see Sec. 4.3) and evaluation (see Sec. 4.4) to optimize its parameters and evaluate its end-to-end solver's performance, thereby populating a database of program-performance pairs. The LLM then acts as an evolution engine: it samples programs from this

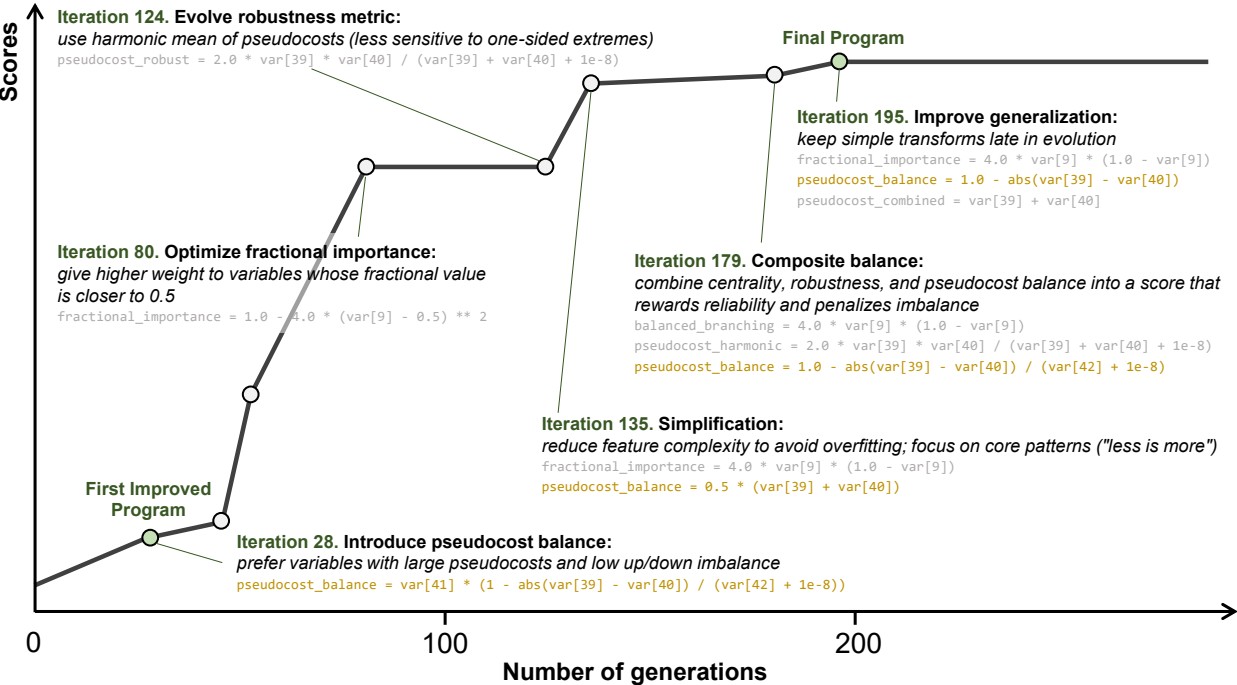

*Figure 2.* Evolutionary process of the branching policy in LLM4Branch. The curve tracks the solver's performance improvement over 200 generations. We highlight the key evolutionary milestones, and provide code-level analyses of these corresponding program updates (see Appendix E for full details).

database to generate a new program, composing a program skeleton $s$ and its associated parameters $\boldsymbol{\theta}^0$. Figure 2 illustrates a example of the evolution process.

### 4.3. Parameter Optimization for Branching Policy

In the newly generated program, the LLM may yield poor parameters $\boldsymbol{\theta}^0$. Since these parameters critically affect branching performance, parameter optimization is essential. However, this is challenging due to the solver's black-box nature and high execution cost. To overcome these challenges, we employ a two-stage process that first rapidly filters underperforming programs, then optimizes the parameters of the promising programs using zeroth-order optimization.

The process begins with the program $p = (s, \boldsymbol{\theta}^0)$ where $\boldsymbol{\theta}^0$ is LLM-generated parameters for skeleton $s$. This program is tested on a small training subset $\mathcal{D}_{\text{sub}} \subseteq \mathcal{D}$, and its performance is benchmarked against a solver internal baseline. Candidates that fail to meet a predefined performance threshold are immediately discarded, thereby conserving computational resources for more promising programs.

Programs that pass the above filter advance to the second stage, where we seek the optimal parameter vector $\boldsymbol{\theta}^*$ for a given skeleton $s$. Given that the solver cost is a black-box function with respect to $\boldsymbol{\theta}$ and provides no gradient information, we employ the zeroth-order optimization algorithm to iteratively probes the solver with different parameter vec-

tors, using the resulting solver cost on the training subset as feedback to find parameters $\boldsymbol{\theta}^*$ with the minimal cost for that specific skeleton.

### 4.4. Branching Policy Evaluation

Upon obtaining the optimized program $p^* = (s, \boldsymbol{\theta}^*)$, we evaluate its performance on the full dataset $\mathcal{D}$. We integrate the policy $p^*$ into the MILP solver and execute it on all instances $d \in \mathcal{D}$, recording the aggregated performance metrics. The resulting cost $M(p^*; \mathcal{D})$, paired with the program $p^*$, is then stored in the historical database. In contrast to the parameter optimization phase, which utilizes a subset for computational efficiency, this evaluation can validate the policy's ability to generalize to a broader instance set.

## 5. Experiments

### 5.1. Experimental Settings

**Benchmark** We evaluated our method on four standard and two challenging MILP problem benchmarks. The four standard ones include set covering (Balas & Ho, 2009), combinatorial auction (Leyton-Brown et al., 2000), capacitated facility location (Cornuéjols et al., 1991), maximum independent set (Bergman et al., 2016)), and the two challenging ones include balanced item placement (Gasse et al., 2022a) and neural network verification (Nair et al., 2020). For con-

*Table 1.* Performance of branching polices on standard benchmarks. We report solving times, number of times a method won (in solving time) over total finished runs, and number of nodes. To maintain a fair comparison, 'Wins' are calculated exclusively among CPU-based methods; GPU-accelerated models (e.g., SORREL, GNN-GPU) serve as reference points only. For hard benchmarks, node comparison may not fully reflect efficiency, as the solver frequently reaches the time limit on these instances. The best performing results are in bold.

| Setcover: | Easy | | | Medium | | | Hard | | |
|---|---|---|---|---|---|---|---|---|---|
| Method | Time↓ | Nodes↓ | Wins↑ | Time↓ | Nodes↓ | Wins↑ | Time↓ | Nodes↓ | Wins↑ |
| FSB | 13.47 | 16.64 | 0/80 | 417.62 | 246.80 | 0/71 | 3159.86 | 863.27 | 0/8 |
| RPB | 5.67 | **55.16** | 0/80 | 53.83 | 2474.10 | 1/80 | 982.84 | 75384.37 | 29/60 |
| MLP | 3.45 | 164.39 | 5/80 | 46.31 | 3133.68 | 4/80 | 1539.34 | 84732.80 | 0/49 |
| Hybrid | 3.30 | 146.21 | 26/80 | 40.18 | 2410.67 | 20/80 | 1044.42 | 60894.14 | 2/60 |
| GNN | 4.23 | 135.96 | 0/80 | 75.44 | **2081.28** | 0/80 | 1832.61 | **30551.97** | 0/40 |
| Symb4CO | 3.37 | 168.77 | 6/80 | 45.50 | 2995.42 | 1/80 | 1086.45 | 74942.62 | 0/59 |
| LLM4Branch | **3.21** | 166.49 | **43/80** | **37.64** | 2724.87 | **54/80** | **952.92** | 64479.94 | **30/61** |
| SORREL | 3.53 | 174.30 | -/80 | 57.47 | 3575.97 | -/80 | 1172.81 | 67181.72 | -/55 |
| GNN-GPU | 3.25 | 135.96 | -/80 | 34.38 | 2081.29 | -/80 | 847.09 | 48248.97 | -/63 |

| Cauctions: | Easy | | | Medium | | | Hard | | |
|---|---|---|---|---|---|---|---|---|---|
| Method | Time↓ | Nodes↓ | Wins↑ | Time↓ | Nodes↓ | Wins↑ | Time↓ | Nodes↓ | Wins↑ |
| FSB | 3.36 | 7.41 | 0/80 | 165.66 | 133.69 | 0/80 | 2661.03 | 739.16 | 0/31 |
| RPB | 2.05 | **12.47** | 0/80 | 21.59 | 1407.88 | 1/80 | **221.59** | **17125.22** | **33/80** |
| MLP | 1.17 | 79.06 | 17/80 | 15.88 | 1914.63 | 1/80 | 469.95 | 57463.13 | 0/75 |
| Hybrid | 1.20 | 72.51 | 8/80 | 14.38 | 1454.31 | 10/80 | 259.56 | 23523.30 | 3/80 |
| GNN | 1.32 | 70.31 | 6/80 | 17.78 | **1293.94** | 0/80 | 229.19 | 17151.55 | 22/80 |
| Symb4CO | 1.19 | 85.41 | 10/80 | 15.12 | 1843.77 | 15/80 | 303.47 | 30099.80 | 1/78 |
| LLM4Branch | **1.09** | 76.17 | **39/80** | **13.12** | 1647.53 | **53/80** | 244.63 | 26901.00 | 21/79 |
| SORREL | 1.27 | 79.28 | -/80 | 15.62 | 1892.73 | -/80 | 204.33 | 22625.49 | -/80 |
| GNN-GPU | 1.18 | 70.31 | -/80 | 12.92 | 1293.94 | -/80 | 199.01 | 17151.55 | -/80 |

| Facilities: | Easy | | | Medium | | | Hard | | |
|---|---|---|---|---|---|---|---|---|---|
| Method | Time↓ | Nodes↓ | Wins↑ | Time↓ | Nodes↓ | Wins↑ | Time↓ | Nodes↓ | Wins↑ |
| FSB | 55.69 | 35.88 | 1/80 | 344.53 | 76.67 | 0/80 | 2305.51 | 41.40 | 0/43 |
| RPB | 42.07 | **68.20** | 1/80 | 181.47 | **148.27** | 0/80 | 674.82 | **182.23** | 2/80 |
| MLP | 31.36 | 220.27 | 13/80 | 122.58 | 336.44 | 14/80 | 533.71 | 410.78 | 23/80 |
| Hybrid | 33.57 | 212.30 | 6/80 | 123.50 | 351.23 | 21/80 | 602.82 | 437.94 | 5/79 |
| GNN | 37.32 | 196.32 | 13/80 | 139.85 | 342.93 | 6/80 | 730.12 | 444.55 | 3/78 |
| Symb4CO | 35.33 | 251.99 | 10/80 | 142.33 | 371.94 | 3/80 | 590.56 | 434.51 | 12/80 |
| LLM4Branch | **30.07** | 234.73 | **36/80** | **117.94** | 353.36 | **36/80** | **512.31** | 405.74 | **30/80** |
| SORREL | 30.96 | 223.03 | -/80 | 124.66 | 375.66 | -/80 | 726.18 | 476.19 | -/80 |
| GNN-GPU | 28.15 | 196.32 | -/80 | 120.67 | 342.93 | -/80 | 609.39 | 447.79 | -/80 |

| Indset: | Easy | | | Medium | | | Hard | | |
|---|---|---|---|---|---|---|---|---|---|
| Method | Time↓ | Nodes↓ | Wins↑ | Time↓ | Nodes↓ | Wins↑ | Time↓ | Nodes↓ | Wins↑ |
| FSB | 360.22 | 52.34 | 0/80 | 2331.43 | 166.2 | 0/40 | 3598.49 | 69.30 | 0/1 |
| RPB | 24.48 | 495.27 | 1/80 | 146.77 | 5551.77 | 5/80 | 2344.12 | **65811.41** | 1/27 |
| MLP | 16.56 | 618.15 | 4/80 | 156.79 | 7661.89 | 0/73 | 2963.66 | 81800.76 | 0/13 |
| Hybrid | 12.60 | 406.31 | 21/80 | 112.48 | 4789.88 | 5/75 | 2656.07 | 77861.76 | 0/16 |
| GNN | 13.60 | **379.97** | 12/80 | 118.63 | 4864.39 | 4/75 | 2823.84 | 75748.50 | 0/15 |
| Symb4CO | 32.88 | 1285.59 | 0/80 | 521.35 | 22344.50 | 0/66 | 3451.28 | 94380.37 | 0/2 |
| LLM4Branch | **10.91** | 453.86 | **42/80** | **62.45** | **3276.22** | **66/80** | **1512.33** | 68606.59 | **38/39** |
| SORREL | 16.63 | 673.47 | -/80 | 100.31 | 5728.51 | -/80 | 2133.85 | 75195.04 | -/21 |
| GNN-GPU | 11.80 | 379.97 | -/80 | 110.88 | 4864.39 | -/75 | 2756.60 | 89034.27 | -/15 |

venience, we refer to these datasets as Setcover, Cauctions, Facilities, Indset, Item Placement, and NNVerify respectively. Following previous works (Gasse et al., 2019; Gupta et al., 2020; Kuang et al., 2024), we generate instances for the four standard benchmarks and categorize them into three difficulty levels: Easy, Medium, and Hard. Specifically, the

*Table 2.* Performance comparison of branching policies on two challenging benchmarks. We report the average solving time, number of times a method won, and the number of nodes. For the Item Placement benchmark, which cannot be solved within the one-hour time limit, we report the primal-dual gap instead of solving time. Note that node comparison in this benchmark may not fully reflect efficiency. The best performing results are highlighted in bold.

| | Item Placement | | | NNVerify | | |
|---|---|---|---|---|---|---|
| Method | Gap↓ | Nodes↓ | Wins↑ | Time↓ | Nodes↓ | Wins↑ |
| FSB | 0.8297 | 24108.97 | 11/0 | 281.20 | 617.64 | 0/80 |
| RPB | 0.8123 | 820584.84 | 5/0 | 66.29 | 1744.58 | 4/72 |
| MLP | 0.8465 | 789629.42 | 2/0 | 37.26 | **829.01** | 21/80 |
| Hybrid | 0.8577 | 619934.66 | 0/0 | 84.33 | 2109.75 | 11/65 |
| GNN | 0.8251 | **137032.23** | 5/0 | 265.64 | 1217.80 | 0/68 |
| Symb4CO | 0.8581 | 841906.43 | 0/0 | 68.69 | 1809.30 | 15/80 |
| LLM4Branch | **0.7196** | 817523.39 | **57**/0 | **34.54** | 1014.08 | **29/80** |
| GNN-GPU | 0.8073 | 419669.96 | -/0 | 44.82 | 1716.00 | -/75 |

Easy set consists of small instances used for training and testing, while the Medium and Hard sets contain larger instances designed to evaluate generalization performance. Detailed benchmark sizes and hyperparameters are reported in Table 9 in Appendix C.

**Baselines** There are seven baselines to compare in this section to illustrate the performance of our method. Specifically, we include reliability pseudocost branching (RPB) and full strong branching (FSB), which are human-designed policies. For machine learning approaches, we evaluate two graph neural networks: the advanced GNN (Gasse et al., 2019) trained via imitation learning and SOEEL (Feng & Yang, 2025) trained via reinforcement learning. We also include Hybrid (Gupta et al., 2020) as a strong CPU-based baseline. Additionally, we include an MLP policy trained on the same input features as our method. This provides a direct comparison between neural network model and our discovered policy given the same features. Finally, we compare against Symb4CO (Kuang et al., 2024), a symbolic regression method that is highly relevant as it also utilizes the same input features to generate CPU-friendly policies.

**Training Settings** We use only *eight* per benchmark for the evolutionary search. Specifically, a subset $\mathcal{D}_{sub}$ of four instances is used for parameter optimization, while the full set $\mathcal{D}$ (comprising all eight instances) is used for policy evaluation. The evolutionary framework is powered by the DeepSeek-R1 (Guo et al., 2025) and runs for 200 main iterations. Within the parameter optimization loop, we perform 50 iterations using Bayesian optimization implemented via the `Scikit-optimize` library (Head et al., 2021). Specifically, the end-to-end optimization objective is designed to minimize the geometric mean number of branching nodes across the training instances, a metric chosen for its deterministic nature and hardware independence. However, for the Item Placement benchmark, where solving to optimality is time-consuming, we instead minimize the geometric

mean of the primal-dual gap within a 180-second time limit. For the fast check of parameter optimization, we retain programs only if its worst-case runtime over the instances does not exceed 125% of node count the solver's default RPB policy. We also normalize input features of all candidates inside each B&B node to $[0, 1]$, following previous work (Gupta et al., 2020). For all the learning-based baselines, we use their official implementations and the default settings. More training details for each learning-based baseline can be found in the Appendix E.

**Evaluation Settings** For four standard benchmarks, we use 80 instances for each difficulty level to evaluate the performance and the generalization ability. Similarly, for two challenging benchmarks, we sample 80 instances directly from their official test sets. We use SCIP 8.0.0 (Achterberg, 2009) as the underlying solver, replacing its default branching policy with the policy under evaluation. All evaluations are conducted with a one-hour time limit per instance. We report standard metrics from the MILP community for benchmarking B&B solvers: (a) Time: the 1-shifted geometric mean (Achterberg, 2009) of running time in seconds, including unsolved instances; (b) Nodes: the 1-shifted geometric mean of B&B nodes generated by each policy, a hardware-independent measure; (c) Wins: number of times that a branching policy wins all the others over total number of solved instances.

## 5.2. Main Results

We compare LLM4Branch to all baselines, with detailed results presented in Table 1, 2. The results show that our discovered policies, which are purely CPU-based, achieve performance highly competitive with the GPU-based GNN policy and outperforms all other CPU-based baselines. Specifically, for large-scale Item Placement instances where solvers reach time limits, our policies is able to close a larger optimality gap than other branching baselines.

Furthermore, our discovered policies exhibit strong generalization to harder instances. We also observe that policies trained on one benchmark tend to generalize well to unseen benchmarks (see Table 10 in Appendix G). Notably, our method achieves such performance using only eight training instances and without requiring any expert demonstrations.

We further analyze whether our policy decision align with the strong branching expert, which is the training target for many imitation learning methods. As shown in Table 3, our discovered policy deviates from the expert's decisions more often than the baselines. Notably, on Indset problems, our policy aligns with the expert in only $24.72\%$ of the cases. Despite this, our policy significantly outperforms all baselines on medium and hard Indset instances (see Table 1). This suggests that our method successfully discovers novel heuristics that innovate beyond the policies imitating strong branching.

*Table 3.* Test accuracy (%) with strong branching expert. For each benchmark, we evaluated the accuracy using 20,000 strong branching samples collected from 2,000 Easy instances.

| Policy | Setcover | Cautions | Facilities | Indset |
|---|---|---|---|---|
| Symb4CO | 55.21 | 52.88 | 65.92 | 41.96 |
| MLP | 55.31 | 55.24 | 69.19 | 37.79 |
| Hybrid | 58.04 | 55.47 | 69.54 | 46.57 |
| GNN | 62.91 | 57.29 | 71.85 | 48.84 |
| LLM4Branch | 49.68 | 43.14 | 61.55 | 24.72 |

We observe that the generated policies are human-readable with explicit semantic definitions and comments. The design of these policies reflects the intuition underlying human-designed heuristics (see Appendix G). We believe that LLM4Branch allows researchers to further understand and optimize these discovered policies.

### 5.3. Ablation Study

To validate the key design choices of our method, we conduct a series of ablation studies on the Setcover and Indset datasets.

**Impact of the Optimization Objective**  A central claim of our work is that optimizing a surrogate objective does not reliably translate into superior end-to-end solver performance. To verify this, we create a variant, **LLM4Branch-IL**, where the solver cost feedback in the evolutionary loop is replaced by the accuracy of predicting strong branching decisions. As shown in Table 4, although LLM4Branch-IL achieves higher imitation accuracy, it fails to yield competitive solving efficiency. In contrast, our default method significantly outperforms this variant in both time and node counts, confirming the necessity of directly minimizing the solver cost.

*Table 4.* Ablation study on the optimization objective.

| Setcover | Time | Nodes | Accuracy (%) |
|---|---|---|---|
| LLM4Branch | **3.21** | **166.49** | 49.68 |
| LLM4Branch-IL | 3.88 | 202.53 | **53.76** |

| Indset | Time | Nodes | Accuracy (%) |
|---|---|---|---|
| LLM4Branch | **10.91** | **453.86** | 24.72 |
| LLM4Branch-IL | 45.21 | 1919.40 | **39.44** |

**Impact of Core Components**  We analyze the contribution of two core components: the parameter optimization and the feature description provided to the LLM. As detailed in Table 5, we compare our full method against two ablated versions. **w/o Param-Opt** removes the parameter optimization stage, forcing the LLM to generate the entire program $p$, including its parameters $\theta$. **w/o Prior** removes the natural language description of variable features from the prompt. The results show a clear performance drop for both variants across all metrics, confirming that parameter optimization and feature description are essential for generating efficient branching policies.

*Table 5.* Ablation study of key algorithm components.

| | Setcover | | Indset | |
|---|---|---|---|---|
| | Time | Nodes | Time | Nodes |
| LLM4Branch | **3.21** | **166.49** | **10.91** | **453.86** |
| w/o Param-Opt | 4.53 | 222.35 | 28.09 | 1336.67 |
| w/o Prior | 4.10 | 209.61 | 17.03 | 827.36 |

**Impact of Different LLM Backbones**  We conduct experiments across a diverse set of LLMs, including GPT-5, DeepSeek-V3, and Qwen3-Next-80B-A3B. As shown in Table 6, the performance remains consistently high across all tested backbones. These results suggest that the proposed method demonstrates strong robustness, achieving effective results across a wide range of models, including both mid-sized and general-purpose LLMs.

*Table 6.* Performance comparison across different LLM backbones.

| | Setcover | | Indset | |
|---|---|---|---|---|
| Backbone LLM | Time | Nodes | Time | Nodes |
| Deepseek-R1 | 3.21 | 166.49 | 10.91 | 453.86 |
| GPT-5 | 3.25 | 169.91 | 9.01 | 391.73 |
| DeepSeek-V3 | 3.39 | 171.99 | 9.34 | 422.71 |
| Qwen3-Next-80B-A3B | 3.55 | 182.72 | 10.52 | 447.12 |

# 6. Conclusions

In this work, we presented LLM4Branch, an automated framework for leveraging LLMs to discover efficient branching policies in MILP solvers. Our results show that LLM4Branch successfully discovers efficient branching policy that generalize well from small training instances to larger instances. Notably, the discovered policies achieve SOTA performance among CPU-based methods and remain highly competitive with advanced GPU-based baselines. Future work will focus on extending this framework to other solver components, such as node selection and cut generation, to further enhance solver efficiency.

# Acknowledgements

This work was supported by National Natural Science Foundation of China (62325305, U25A20462), the Research and Development Project of CRSC Research & Design Institute Group Co., Ltd. and BNRist project (No. BNR2024TD03003).

# Impact Statement

This paper presents work whose goal is to advance the field of Machine Learning. There are many potential societal consequences of our work, none which we feel must be specifically highlighted here.

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

# A. The Gap between IL Accuracy and End-to-end Performance

To investigate the gap between the surrogate objective and practical solver performance, we trained an MLP-based branching policy on the Indset dataset. The problem instances were configured with 500 nodes and an affinity of 4. For training, we generated 100,000 samples from 10,000 instances. For validation, we employed 20,000 samples from a separate set of 2,000 instances to evaluate prediction accuracy, while using 80 instances to measure the actual solving time. We performed 10 independent runs to ensure robust findings.

We first compare checkpoints extracted based on different performance metric. Specifically, from each run, we identified three distinct checkpoints: the **Loss-optimal** checkpoint (lowest validation loss), the **Accuracy-optimal** checkpoint (highest validation accuracy), and the **Time-optimal** checkpoint (shortest solving time on the 80 validation instances). As summarized in Table 7, checkpoints achieving the best surrogate metrics (Loss-optimal and Accuracy-optimal) fail to deliver the fastest solving speed. In contrast, the Time-optimal checkpoints yield superior end-to-end performance, despite exhibiting higher loss and lower accuracy. This comparison provides direct evidence that high prediction accuracy does not strictly correspond to efficient solving.

*Table 7.* The performance of checkpoints under different validation metric. Checkpoints optimized for direct solver efficiency (Time-optimal) achieve better solving performance, despite higher surrogate loss.

| Selected Checkpoint | Loss ↓ | Accuracy (%) ↑ | Solve Time (s) ↓ |
|---|---|---|---|
| Loss-Optimal | **3.057 ± 0.002** | 44.02 ±0.16 | 3.047 ± 0.072 |
| Accuracy-Optimal | 3.160 ± 0.039 | **47.47 ± 0.33** | 3.044 ± 0.059 |
| Time-Optimal | 3.094 ± 0.023 | 43.84 ± 1.02 | **3.016 ± 0.058** |

We further observe this gap in the training dynamics shown in Figure 3. While the validation loss (blue curve) decreases steadily and becomes stable, the solving time (orange curve) shows a different trend. After an initial improvement, the solving time stagnates around epoch 100 and starts to fluctuate, even though the loss continues to decrease. Moreover, the large variance (shaded orange area) indicates that models with similar low loss values can result in very different solving speeds across different runs. This confirms that simply minimizing the loss function does not guarantee a continuous improvement in the actual solving time.

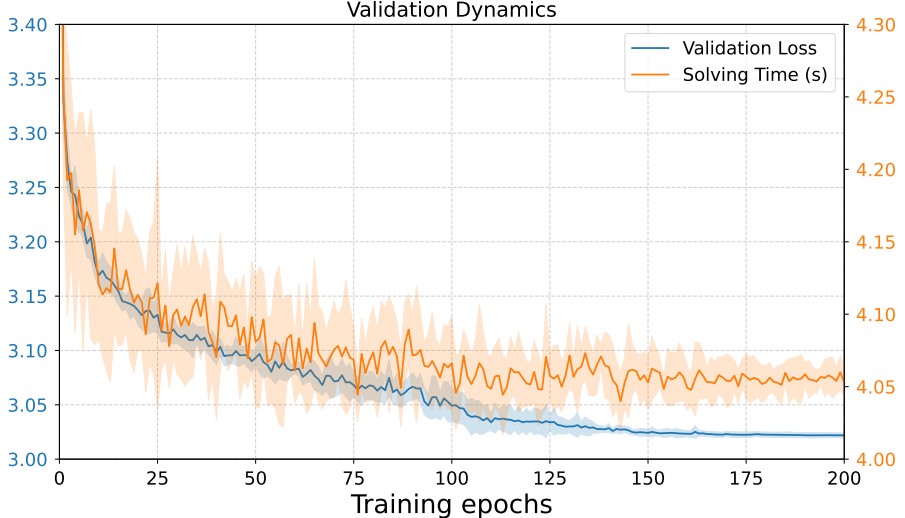

*Figure 3.* **Training dynamics of the MLP policy.** The blue curve represents the validation loss, while the orange curve shows the average solving time on the 80 validation instances (shaded areas indicate standard deviation across 10 runs). Although the loss consistently decreases, the solving time stagnates after epoch 100 and exhibits high variance, illustrating the gap between the surrogate objective and actual solver efficiency.

## B. Input Features

We adopt the 91–dimensional feature set following the previous works (Gupta et al., 2020; Kuang et al., 2024) as the input to our branching policy. These features are broadly categorized into static features, which describe the MILP instance itself, and dynamic features, which reflect the solver's current state. A complete listing is provided below.

*Table 8.* Feature description for MILP branching variable proposed in (Khalil et al., 2016; Gupta et al., 2020)

| Index | Name | Description |
|---|---|---|
| **Part 1: NodeBipartite Variable Features (indices 0-18)** | | |
| 0 | Objective coefficient | Objective coefficient of the variable |
| 1-4 | Variable type | Binary indicators: binary, integer, implicit integer, continuous |
| 5-6 | Bound indicators | Whether the variable has lower/upper bound |
| 7 | Normalized reduced cost | Reduced cost normalized by the norm of constraint coefficients |
| 8-9 | LP solution value | LP solution value and its fractional part |
| 10-11 | LP solution bound | Whether the variable reaches its lower/upper bound in current LP solution |
| 12 | Scaled age | Number of LP iterations since last basic, normalized by total LP iterations |
| 13-14 | Incumbent values | Value in current best primal solution and average value in previous feasible solutions |
| 15-18 | Basis status | Indicators for variable status: lower, basic, upper, zero |
| **Part 2: Khalil2016 Features (indices 19-90)** | | |
| 19-21 | Objective coefficients | Raw value, positive part only, negative part only |
| 22 | Number of constraints | Number of constraints that the variable participates in |
| 23-26 | Static constraint degree | Statistics over constraint degrees: mean, standard deviation, minimum, maximum |
| 27-31 | Positive coefficients | Count, mean, standard deviation, minimum, maximum of positive constraint coefficients |
| 32-36 | Negative coefficients | Count, mean, standard deviation, minimum, maximum of negative constraint coefficients |
| 37-38 | Slack and ceil distances | Fractional distance and ceiling distance to nearest integer |
| 39-43 | Pseudocost values | Upwards/downwards pseudocosts, their ratio, sum, and product |
| 44-47 | Infeasibility statistics | Counts and ratios for upwards/downwards cutoffs |
| 48-54 | Dynamic constraint degrees | Dynamic variant of constraint degree statistics and ratios to static counterparts |
| 55-66 | Coefficient ratios | Ratios of constraint coefficients to RHS and one-to-all coefficient interaction ratios |
| 67-90 | Active constraint statistics | Statistics under four weighting schemes (unit, inverse sums, dual cost): count, sum, mean, standard deviation, minimum, maximum |

## C. Benchmark Details

We generate the dataset following the process established in (Gasse et al., 2019), using four NP-hard combinatorial problem benchmarks: set covering, combinatorial auction, capacitated facility location, and maximum independent set. For each benchmark, we create instances at three difficulty levels (Easy, Medium, and Hard) by progressively increasing the problem scale. The specific generation algorithms and corresponding hyperparameters for each benchmark are detailed in Table 9.

For the balanced item placement (Item Placement), we follow the dataset partition established by (Gasse et al., 2022a), which comprises 99,000 training, 1,000 validation, and 100 testing instances. While the baseline methods utilize the full training set, our approach randomly selects only 8 instances from the original training set for policy discovery. For neural network verification (NNVerify) (Nair et al., 2020), we follows the original split of 2,555 training, 549 validation, and 588 testing instances. To ensure a high-quality evaluation, we randomly sample 80 instances from the default test set for our

final evaluation, specifically excluding infeasible instances.

*Table 9.* Instance Generation Algorithms and Parameter Configurations

| Benchmark | Difficulty | Generation Algorithm | Hyperparameters |
|---|---|---|---|
| Set Covering | Easy
Medium
Hard | (Balas & Ho, 2009) | 500 rows, 1000 columns
1000 rows, 1000 columns
2000 rows, 1000 columns |
| Combinatorial Auction | Easy
Medium
Hard | (Leyton-Brown et al., 2000) | 100 items, 500 bids
200 items, 1000 bids
300 items, 1500 bids |
| Capacitated Facility Location | Easy
Medium
Hard | (Cornuéjols et al., 1991) | 100 facilities, 100 customers
100 facilities, 200 customers
100 facilities, 400 customers |
| Maximum Independent Set | Easy
Medium
Hard | (Bergman et al., 2016) | 750 nodes, affinity 4
1000 nodes, affinity 4
1500 nodes, affinity 4 |

## D. Detailed Methodology of the proposed LLM4Branch

### D.1. Prompt of Branching Policy Generation

Our approach utilizes a structured prompt design composed of two complementary components: the System Prompt and the User Prompt. This ensures consistent guidance while allowing dynamic adaptation to specific program contexts.

**System Prompt.** This prompt provides the foundational context and constraints that remain constant across all branching policy generation tasks. It establishes the expert role, defines the problem scope, and specifies the technical requirements for the scoring function.

---

**System Prompt**

You are an expert in machine learning for combinatorial optimization. Design a Python function that scores a candidate variable for branching in a branch and bound algorithm.

The function will receive a feature vector (numpy.ndarray with dimension (n, 91)) including the 91 features of n decision variables and params you extract to use in your score function. You should use these features to compute a score for each variable, return np.ndarray with dimension (n,) Use at most 10 features in total.

Input features (all normalized to [0,1]) are structured as follows:

Part 1 - NodeBipartite Features (indices 0-18): 0: objective coefficient 1-4: binary indicators (binary, integer, implicit integer, continuous) 5-6: bound indicators (has lower/upper bound) 7: normalized reduced cost 8-9: LP solution value and fractional part 10-11: LP solution bound indicators 12: scaled age (iterations since last basic) 13-14: incumbent value and average value 15-18: basis status indicators (lower, basic, upper, zero)

Part 2 - Khalil2016 Features (indices 19-90): 19-21: objective coefficient (raw, positive part, negative part) 22: number of participating constraints 23-26: static constraint degree stats (mean, std, min, max) 27-31: positive constraint coefficients (count, mean, std, min, max) 32-36: negative constraint coefficients (count, mean, std, min, max) 37-38: fractional distance and ceiling distance 39-43: pseudocost values (up, down, ratio, sum, product) 44-47: cutoff counts and ratios (up/down) 48-54: dynamic constraint degrees (mean, std, min, max) and ratios to static (mean, min, max) 55-66: coefficient-to-RHS ratios and coefficient interaction ratios (min/max for positive/negative cases) 67-90: active constraint statistics under four weighting schemes (unit, 1/sum_all, 1/sum_candidate, dual_cost), each including count, sum, mean, std, min, max.

---

**User Prompt.** This prompt introduces dynamic, context-specific information that evolves during the optimization process. It facilitates iterative improvement by providing current performance feedback and inspirational examples. The curly braces {} in the prompts serve as placeholder notation that gets dynamically populated with actual content during execution:

- **{metrics}**: Current performance evaluation metrics of the current branching policy

- **{current_program}**: The Python code implementation of the current branching policy being optimized

- **{inspiration_programs}**: A collection of high-performing branching policies from previous iterations

---

### User Prompt

\# Current Program Information
- Current performance metrics: {metrics}
``` python
{current_program}
```

\# Inspiration Programs
These programs represent diverse approaches and creative solutions that may inspire new ideas:
{inspiration_programs}

\# Task
Rewrite the score function in the program to improve its performance on the specified metrics. Provide the complete new program code. Make sure your rewritten program maintains the same inputs and outputs as the original program, but with improved internal implementation.
IMPORTANT: In you code, you must write three line at the front with format:
1. "USED_FEATURES = []" (list, including the indices you used in score function, As for unused features, you must exclude them from USED_FEATURES.)
2. "PARAMS = []" (list, extract the hyperparameter to this list)
3. "BOUNDS = [[]]" (list[list], the bound of all hyperparameter, for example, BOUNDS = [[0, 1], [-1, 3]], the first parameter has min 0, and max1, the second parameter has min -1, max 3. Attention! the list must have the format as [[0, 1], [-1, 3]]. Format as [0, 1] * 10 is not allowed)

``` python
# Your rewritten score function here
```

---

### D.2. Pseudocode of the Proposed LLM4Branch

Algorithm 1 outlines the complete LLM4Branch framework, integrating program generation, parameter optimization, and policy evaluation into an evolutionary process. The following paragraphs detail key components of the algorithm, with additional implementation specifics provided in subsequent algorithms.

**Initial Program** The algorithm begins with a simple random branching policy that serves as the baseline for evolutionary improvement. This initial program selects branching variable uniformly at random from the candidate variables, providing a feasible starting point for the optimization process. The corresponding program skeleton is implemented as follows:

```python
import random
import numpy as np

USED_FEATURES = [1]
PARAMS = [0.5]
BOUNDS = [[0, 1]]
def score_function(variable_features, params):
    return np.random.rand(variable_features.shape[0]) * params[0]
```

**Program Sample Strategy**    At the evolutionary iteration, one candidate program is sampled from program database $\mathcal{P}$ using a fixed probability to choose between exploration and exploitation. Exploration involves randomly sampling programs from the entire database to discover novel program structures and prevent premature convergence to local optima. Exploitation prioritizes programs with higher performance scores to refine and build upon successful branching strategies.

**Inspiration Sample Strategy**    We adopt an island-based management strategy following (Novikov et al., 2025), grouping programs in $\mathcal{P}$ by structural and feature-based characteristics. For a candidate $p_0$, inspiration programs are sampled in its island, including both top-$k$ performers and others that are structurally diverse, with diversity quantified by edit distance and feature dissimilarity.

---

**Algorithm 1** Pseudocode of the LLM4Branch

---

**Require:** Training instance set $\mathcal{D}$, Training subset $\mathcal{D}_{\text{sub}} \subseteq \mathcal{D}$, LLM model $\pi$, performance threshold $\tau$, iteration limit $T$
 1: Initialize program database $\mathcal{P}$ with initial program
 2: **for** $t = 1$ **to** $T$ **do**
 3:      // Sample candidate programs from database
 4:      Sample one candidate program $p_0$ from database $\mathcal{P}$
 5:      Sample some inspiration programs $p_1, p_2, ..., p_n$ from database $\mathcal{P}$
 6:      // Generate program based on LLM
 7:      Build the prompt $o$ based on the current program $p_0$ and inspirations $p_1, p_2, ..., p_n$
 8:      Sample new program skeleton and its parameters from LLM $(s, \boldsymbol{\theta}^0) \sim \pi(\cdot|o)$
 9:      // Fast filtering
10:      perf $\leftarrow$ EVALUATEONSUBSET$((s, \boldsymbol{\theta}^0), \mathcal{D}_{\text{sub}})$
11:      **if** perf $< \tau$ **then**
12:          **continue** // Discard underperforming skeleton
13:      **end if**
14:      // Parameter optimization
15:      $\boldsymbol{\theta}^* \leftarrow$ PARAMETEROPTIMIZION$(s, \boldsymbol{\theta}^0, \mathcal{D}_{\text{sub}})$
16:      $p^* \leftarrow (s, \boldsymbol{\theta}^*)$
17:      // Policy evaluation
18:      cost $\leftarrow M(p^*, \mathcal{D})$
19:      $\mathcal{P} \leftarrow \mathcal{P} \cup \{(p^*, \text{cost})\}$
20: **end for**
21: **return** the program $p_{\text{opt}} = (s_{\text{opt}}, \boldsymbol{\theta}_{\text{opt}})$ with the lowest cost in the database $\mathcal{P}$

---

# E. More Experiment Details

**Implementation Details**    We build upon the OpenEvolve 0.1.0 (Sharma, 2025) library for evolutionary optimization and utilize Ecole 8.0.1(Prouvost et al., 2020) for data generation and feature extraction. To reduce feature extraction overhead, we modified Ecole's source code to extract only essential features. During policy evaluation and parameter optimization, which typically involves solving multiple MILP instances, we distribute the solving tasks across multiple processes concurrently, thereby significantly reducing the total evaluation time. To ensure accurate computation time measurements, we implement

---

**Algorithm 2** Parameter Optimization

---

**Require:** Program skeleton $s$, initial program parameter $\boldsymbol{\theta}^0$, training instance set $\mathcal{D}$
1: Initialize optimizer from `scikit-optimize` and iteration limit $T_{\text{param}}$ from config file
2: Set $\boldsymbol{\theta} \leftarrow \boldsymbol{\theta}^0$
3: **for** $t = 1$ **to** $T_{\text{param}}$ **do**
4:     Evaluate policy $(s, \boldsymbol{\theta})$ on $\mathcal{D}$: $c \leftarrow M((s, \boldsymbol{\theta}), \mathcal{D})$
5:     Update optimizer with $(\boldsymbol{\theta}, c)$
6:     Sample new parameters $\boldsymbol{\theta}$ from optimizer
7: **end for**
8: **return** the parameters $\boldsymbol{\theta}^*$ with the lowest solver cost

---

CPU affinity binding (core pinning) for each solving task, guaranteeing that each task runs exclusively on a single dedicated core and that the entire solving process remains confined to that core. For all the ML-based baselines, we use their official implementations and the default settings. Specifically, Symb4CO was trained on 1,000 samples from ten instances, while the other ML baselines ware trained on 100,000 samples from 10,000 instances and validated on 20,000 samples from 2,000 instances. For two challenging benchmarks, we adopt the original dataset partitions without modification. All experiments are performed on a machine with an AMD EPYC 7742 64-Core CPU at 2.25GHz and an NVIDIA GeForce RTX 4090 GPU.

**Hyperparameter Setting** We set the evolutionary process to run for 200 iterations, with exploration and exploitation probabilities set to 0.7 and 0.3, respectively. For the large language models, we employ DeepSeek-R1 (Guo et al., 2025), configuring the generation with a temperature of 0.7, top-p of 0.95, and a maximum token limit of 8192. All baseline methods use the hyperparameters specified in their original papers (Gasse et al., 2019; Gupta et al., 2020; Kuang et al., 2024) or official open-source implementations.

**Running Cost** The duration of a single LLM4Branch run can range from approximately six to ten hours, depending on the evaluation runtime, the hardware used and hyperparameter setting. In terms of token usage, each run consumes about 1.2 million tokens, costing approximately \$0.2, based on the pricing of DeepSeek-R1.

# F. Evolution of the Discovered Branching Policy

This section provides a concise analysis of the evolution process of the discovered branching policy of Cauctions (Appendix H.2). The concrete trajectory and implementation details are shown in Figure 4. We discuss representative evolutionary iterations below to demonstrate how the policy improves.

- **Iteration 28. Introduce pseudocost balance.** A new term `pseudocost_balance = var[41] * (1 - abs(var[39] - var[40]) / (var[42] + 1e-8))` was introduced to prefer variables with large pseudocosts while penalizing strong up/down imbalance. This term increases the score of candidates that are both influential (high pseudocost) and reasonably symmetric in their up/down effect, which leads to more reliable branching choices.

- **Iteration 80. Optimize fractional importance.** The policy incorporated a preference for fractional variables via the term `fractional_importance = 1.0 - 4.0 * (var[9] - 0.5) ** 2`. This effectively increases the score for variables near 0.5, aligning with the intuition that such variables offer higher potential for branching improvements.

- **Iteration 124: Update pseudocost combination.** The policy adopted the harmonic mean to combine up and down pseudocosts: `score = 2.0 * var[39] * var[40] / (var[39] + var[40] + 1e-8)`. Unlike arithmetic averaging, this composite metric requires significant gains in both directions to yield a high score, thereby filtering out variables with one-sided potential.

- **Iteration 135. Simplification.** To avoid overfitting, the policy was simplified ("less is more"): fractional importance was expressed as `4.0 * var[9] * (1.0 - var[9])` and pseudocosts were averaged as `pseudocost_balanced = 0.5 * (var[39] + var[40])`. The simplification emphasizes core patterns that generalize better across instances.

- **Iteration 179. Composite balance.** A composite score combined centrality, robustness, and pseudocost balance

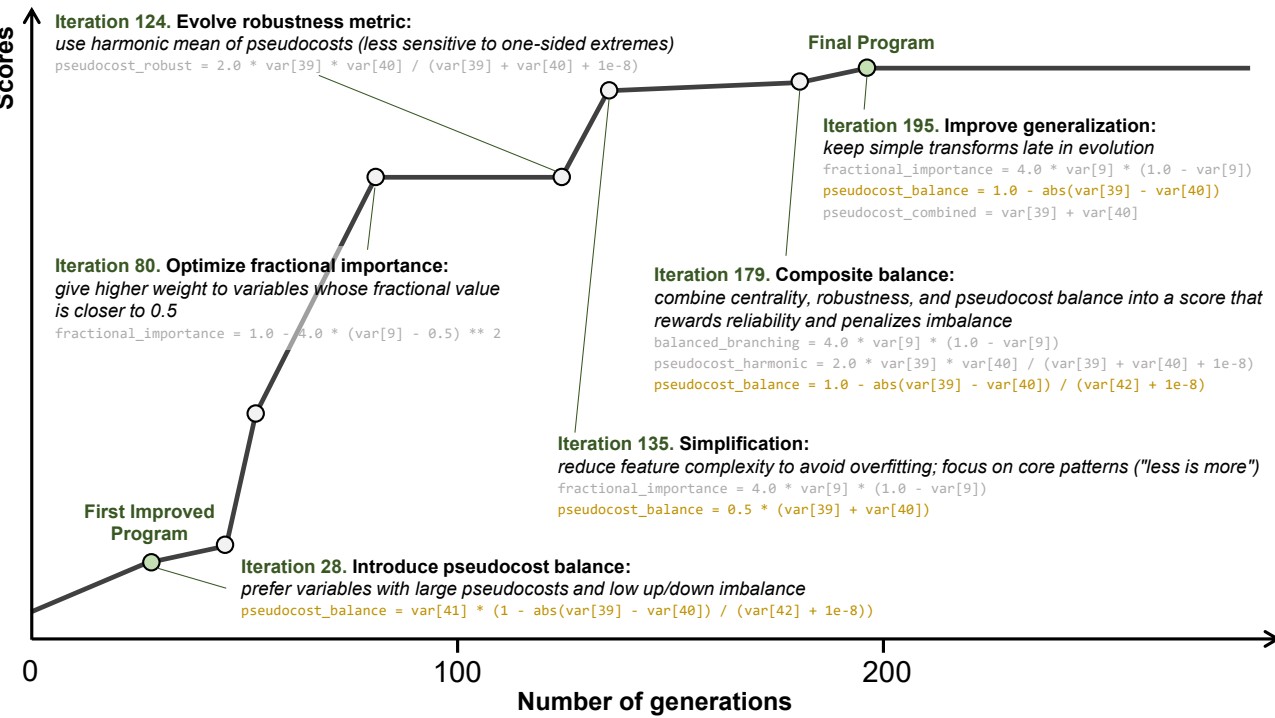

*Figure 4.* Evolutionary process of the branching policy in LLM4Branch. The curve tracks the solver's performance improvement over 200 generations. We highlight the key evolutionary milestones, and provide code-level analyses of these corresponding policy updates.

(`balanced_branching`, `pseudocost_harmonic`, `pseudocost_balance`) to reward reliability while penalizing imbalance. This multi-term composition is more stable and performs well across diverse problem instances.

- **Iteration 195. Improve generalization.** Late-stage evolution preserved several simple transforms to favor broadly applicable rules: e.g., `fractional_importance = 4.0 * var[9] * (1.0 - var[9])`, `pseudocost_balance = 1.0 - abs(var[39] - var[40])`, and `pseudocost_combined = var[39] + var[40]`. The emphasis shifted from highly specific feature combinations to simple, broadly applicable operations, thereby improving generalization. Note that `pseudocost_balance` appears repeatedly across multiple iterations (highlighted in the figure), indicating that this component plays an important role.

In summary, the evolution exhibited a rapid early improvement (introduction of targeted heuristics), followed by refinements and simplification, and finally a plateau where a compact, general policy is retained. The final discovered policy not only enables efficient branching decisions but is also easy to understand, and demonstrates strong generalization.

## G. More Discussions

**Cross Benchmark Generalization.** Unlike learning-based methods that often struggle with out-of-distribution generalization, our discovered policies generalize well to unseen benchmarks (see Table 10), frequently outperforming the default RPB. This generalization capability may stem from the compactness of the branching programs.

**Analysis of Generated Code.** The generated code is not a black-box function but a structured program with explicit logic. We illustrate this using the discovered policy of Set Covering (Appendix H.1) as a representative example. (a) The policy does not learn from scratch but instead leverages prior domain knowledge to draw inspiration from human expert design patterns. For instance, the quadratic term `4.0 * fractional_part * (1.0 - fractional_part)` reflects a classic expert strategy to prioritize variables near 0.5. (b) The LLM generates code with clear semantic definitions and explanatory comments. For example, the term `constraint_activity` is explicitly named to represent a variable's participation in active constraints. Combined with comments like "Enhanced constraint participation", this helps readers understand its meaning. (c) The program use NumPy vectorized operations for efficient computation of nonlinear formulas. For example, it directly

*Table 10.* Generalization evaluation of discovered policies across different benchmarks (Time: s). Bold values indicate performance that surpasses the default RPB policy.

| Benchmark \ Policy From | SetCover | Cauctions | Facilities | Indset | RPB |
|---|---|---|---|---|---|
| Setcover | **3.21** | **5.29** | **5.43** | 6.08 | 5.67 |
| Cauctions | **1.30** | **1.09** | 3.64 | **2.01** | 2.05 |
| Facilities | 58.57 | 75.47 | **30.07** | 75.17 | 42.07 |
| Indset | **21.92** | **20.88** | 69.87 | **10.91** | 24.48 |

utilizes functions like `np.tanh` and `np.log1p` to handle nonlinear transformations of features. (d) Feature combinations are constructed with explicit coefficient placeholders, which facilitates subsequent parameter optimization. For instance, `structural_component` keeps coefficients `params[3]-params[6]` for its subterms. To conclude, these properties both improve solver performance and allow efficient computation. Moreover, the discovered policies are easy to understand and often yield new insights.

**Distinction from General Coding Agents.** While general-purpose coding agents (e.g., Claude Code) have shown remarkable capabilities in software engineering tasks, they typically operate on well-defined programming logic with immediate execution feedback. In contrast, discovering an efficient MILP branching policy involves navigating a "black-box" environment where rewards are highly delayed. These inherent complexities necessitate our customized framework.

**Discussions on Limitations and Future Work.** While our framework demonstrates promising results, we conclude some challenges and corresponding exciting future work:

- **Generalization via Regularization.** A challenge lies in generalizing policies from small, easy training instances to larger, harder ones. Optimizing solely for the end-to-end solver cost on easy problems may induce overfitting, where the policy exploits instance-specific shortcuts that do not scale. To mitigate this, future work could introduce an auxiliary **code complexity metric** as a regularization term. By penalizing programs for excessive length, feature redundancy, or Abstract Syntax Tree (AST) node counts, it may incentivize the generation of concise, interpretable rules which are less prone to overfitting the specific scale of training instances.

- **Data Augmentation via Stochastic Exploration.** Our method currently relies on a small instance set (e.g., 8 instances) for training. To enhance data efficiency in this regime, we could integrate stochastic exploration during the evaluation of candidate policies. Instead of deterministically selecting the variable with the highest score, the solver can take a random action with a small probability (e.g., 10%) at each node. This mechanism forces the solver to traverse diverse search trajectories within the same instance, effectively augmenting the training data by exposing the branching policy to a wider variety of solver states without expanding the instance set.

- **Structural Input Representations.** Currently, our policy utilizes a lightweight 91-dimensional feature vector representing branching candidates. While efficient, this representation overlook structural information of MILP instance, which may be capture by variable-constraint bipartite graphs (Gasse et al., 2019). Future work will explore extending our framework to process these richer features.

- **The choice of end-to-end performance metrics.** The end-to-end performance metric in our evaluation is the number of branching nodes. While wall-clock solving time is the primary performance metric, it is often volatile and sensitive to system noise and hardware fluctuations. Differently, the number of branching nodes are deterministic but not always perfectly correlated with time efficiency. For example, Strong Branching typically yields a small number of nodes but incurs an unacceptably high computational cost per node. Consequently, the choice of the performance metric remains a critical challenge and a vital direction for future research.

- **Toward Scientific Discovery.** A challenge in combinatorial optimization is that providing formal theoretical performance guarantees for branching policies remains extremely difficult due to the NP-hard nature of the problem. To move toward deeper scientific discovery, future work could explore the integration of formal verification environments, such as Lean, into the evolutionary loop. By extracting and verifying tractable properties or mathematical invariants during the discovery process, the framework could potentially provide feedback that guides the LLM toward policies that are not only empirically efficient but also theoretically grounded.

# H. Evolutionary Result of LLM4Branch

## H.1. Discovered Branching Policy of Set Covering

```python
USED_FEATURES = [0, 7, 9, 43, 48, 67]
PARAMS = [1.0,0.5887010792086566,0.31746091541407373,0.9398783973248053,0.6031768166959277,
          0.2645470326979516, 0.6762821726601128]

import numpy as np

def score_function(variable_features, params):
    # Extract core features with semantic clarity
    obj_coef = variable_features[:, 0]                   # Objective coefficient
    reduced_cost = variable_features[:, 7]               # Reduced cost (dual information)
    fractional_part = variable_features[:, 9]            # Fractional part of LP solution
    pseudocost_product = variable_features[:, 43]        # Pseudocost product (reliability)
    dyn_degree_mean = variable_features[:, 48]           # Dynamic constraint degree mean
    active_constraint_count = variable_features[:, 67]   # Active constraint count
    # Enhanced nonlinear transformations with improved numerical stability
    # Strong branching centrality with quadratic emphasis on 0.5 fractionality
    centrality = 4.0 * fractional_part * (1.0 - fractional_part)
    # Enhanced constraint participation with proper scaling and saturation
    constraint_activity = np.tanh(dyn_degree_mean * 0.4) * np.tanh(active_constraint_count * 0.06)
    # Strategic feature interactions with better normalization
    obj_fraction_synergy = np.tanh(np.abs(obj_coef)) * centrality
    # Dual information with controlled saturation
    dual_importance = np.tanh(np.abs(reduced_cost) * 2.5)
    # Pseudocost strength with logarithmic scaling for stability
    pseudocost_strength = np.log1p(np.abs(pseudocost_product) + 1e-8)
    # Constraint importance with proper scaling
    constraint_importance = np.log1p(active_constraint_count) / np.log(2.0)
    # Optimized scoring with hierarchical structure and clear separation
    fractional_component = (
        params[0] * centrality +                         # Primary: fractionality centrality
        params[1] * fractional_part                      # Secondary: raw fractional value
    )
    pseudocost_component = (
        params[2] * pseudocost_strength
    )
    structural_component = (
        params[3] * constraint_activity +                # Constraint participation
        params[4] * obj_fraction_synergy +               # Objective-fraction synergy
        params[5] * dual_importance +                    # Dual information
        params[6] * constraint_importance                # Scaled constraint importance
    )
    # Combined score with clear component separation
    raw_score = fractional_component + pseudocost_component + structural_component
    # Improved output transformation using centered softplus for better gradient flow
    return np.log1p(np.exp(raw_score - 2.0))             # Centered softplus for stable outputs
```

## H.2. Discovered Branching Policy of Combinatorial Auction

```python
USED_FEATURES = [7, 9, 22, 39, 40, 43]
PARAMS = [0.0, 1.0, 1.0, 0.26450634387680155, 0.29613190117158167, 0.0]
BOUNDS = [[0, 1], [0, 1], [0, 1], [0, 1], [0, 1], [0, 1]]

import numpy as np

def score_function(variable_features, params):
    # Extract key features for branching
    reduced_cost = variable_features[:, 7]        # Normalized reduced cost
    fractional_part = variable_features[:, 9]     # Fractional part
    constraint_degree = variable_features[:, 22]  # Number of constraints
    pseudocost_up = variable_features[:, 39]      # Up pseudocost
    pseudocost_down = variable_features[:, 40]    # Down pseudocost
    pseudocost_product = variable_features[:, 43] # Pseudocost product

    # Simple transformations only - maintain good generalization
    fractional_importance = 4.0 * fractional_part * (1.0 - fractional_part)
    pseudocost_balance = 1.0 - np.abs(pseudocost_up - pseudocost_down)
    pseudocost_combined = pseudocost_up + pseudocost_down

    # Linear combination with optimized weights
    score = (params[0] * reduced_cost +
             params[1] * fractional_importance +
             params[2] * constraint_degree +
             params[3] * pseudocost_balance +
             params[4] * pseudocost_combined +
             params[5] * pseudocost_product)

    return score
```

## H.3. Discovered Branching Policy of Capacitated Facility Location

```
USED_FEATURES = [0, 7, 8, 9, 37, 39, 40, 43, 44, 45]
PARAMS = [0.04705795345254364, 0.6785051301493801, 0.2969679650822195, 0.6343660825577095,
          0.48452965740137316, 1.0, 0.33402312976578125, 0.16108144646419523, 0.38212310014780954,
          0.40656551874413094]

import numpy as np

def score_function(variable_features, params):
    # Extract core branching features
    obj_coef = variable_features[:, 0]          # objective coefficient
    reduced_cost = variable_features[:, 7]      # reduced cost
    lp_value = variable_features[:, 8]          # LP solution value
    fractional = variable_features[:, 9]        # fractional part
    frac_distance = variable_features[:, 37]    # fractional distance
    pseudo_up = variable_features[:, 39]        # pseudocost up
    pseudo_down = variable_features[:, 40]      # pseudocost down
    pseudo_product = variable_features[:, 43]   # pseudocost product
    cutoff_up = variable_features[:, 44]        # cutoff up count
    cutoff_down = variable_features[:, 45]      # cutoff down count

    # Core branching indicators with robust transformations
    centrality = 4.0 * fractional * (1.0 - fractional)  # Strong centrality measure
    pseudo_reliability = np.minimum(pseudo_up, pseudo_down) / (pseudo_up + pseudo_down + 1e-8)
    bound_tightening = np.log1p(cutoff_up + cutoff_down)

    # Strategic feature interactions
    obj_centrality = obj_coef * centrality
    reliability_tightening = pseudo_reliability * bound_tightening
    reduced_centrality = reduced_cost * centrality
    pseudo_robust = np.sqrt(pseudo_product + 1e-8)

    # Additional strategic components
    distance_penalty = np.sqrt(frac_distance + 1e-8)
    cutoff_asymmetry = np.tanh(cutoff_up / (cutoff_down + 1e-8) - 1.0)
    lp_magnitude = np.abs(lp_value)
    pseudo_sum = pseudo_up + pseudo_down

    # Combined scoring with balanced interactions
    scores = (
        params[0] * centrality + params[1] * pseudo_robust +
        params[2] * pseudo_reliability + params[3] * obj_centrality +
        params[4] * reliability_tightening + params[5] * reduced_centrality +
        params[6] * distance_penalty + params[7] * cutoff_asymmetry +
        params[8] * lp_magnitude + params[9] * pseudo_sum
    )

    return score
```

**H.4. Discovered Branching Policy of Maximum Independent Set**

```python
USED_FEATURES = [0, 7, 9, 22]
PARAMS = [1.0786976161645492, 0.23806897907821495, 1.2220001238555878, 0.9458262526360364]

def score_function(variable_features, params):
    # Extract essential features for branching decision
    obj_coeff = variable_features[:, 0]         # Objective coefficient
    reduced_cost = variable_features[:, 7]      # Reduced cost
    fractional_part = variable_features[:, 9]   # Fractional part of LP solution
    constraint_count = variable_features[:, 22] # Number of participating constraints

    # Core branching preference - prefer variables near 0.5 (peaks at 0.5)
    branching_preference = 4.0 * fractional_part * (1.0 - fractional_part)

    # Optimized linear combination focusing on most impactful features
    score = (
        params[0] * obj_coeff +
        params[1] * reduced_cost +
        params[2] * branching_preference +
        params[3] * constraint_count
    )

    return score
```

## H.5. Discovered Branching Policy of Item Placement

```python
USED_FEATURES = [41, 42, 43, 77]
PARAMS = [1.0, 0.9143725167306516]
BOUNDS = [[0, 1], [0, 1]]

import numpy as np

def score_function(variable_features, params):
    """
    Compute branching scores using an improved weighted combination of key features.
    Features used:
    41: pseudocost ratio
    42: pseudocost sum
    43: pseudocost product
    77: active constraints (dual weighting)
    """
    # Extract parameters
    w_pseudo, w_dual = params

    # Extract features
    pseudo_ratio = variable_features[:, 41]      # Pseudocost ratio
    pseudo_sum = variable_features[:, 42]        # Pseudocost sum
    pseudo_prod = variable_features[:, 43]       # Pseudocost product
    dual_active = variable_features[:, 77]       # Active constraints (dual weighting)

    # Compute individual score components
    # Pseudocost: combine product and sum with reliability factor
    pseudo_reliability = 1.0 / (1.0 + np.abs(pseudo_ratio - 1.0))
    pseudo_score = pseudo_prod * pseudo_reliability + 0.5 * pseudo_sum

    # Dual activity: variables in many active constraints
    dual_score = dual_active

    # Combine all components with weights
    combined = (
        w_pseudo * pseudo_score +
        w_dual * dual_score
    )

    return combined
```

## H.6. Discovered Branching Policy of Neural Network Verification

```python
USED_FEATURES = [0, 7, 9, 12, 41, 42, 90]
PARAMS = [0.09235624591252971, 0.008705701580663753, 0.38505830924832724]
BOUNDS = [[0, 1], [0, 1], [0, 1]]

import numpy as np

def score_function(variable_features, params):
    """
    High-performance branching score based on the top-performing inspiration (0.5673).
    Key principles:
    1. Strong fractionality preference with quadratic peak at 0.5
    2. Balanced pseudocost reliability using sum, product, and ratio
    3. Constraint tightness from dual-cost weighted constraint statistics
    4. Age-based anti-stalling to avoid outdated variables
    5. Reduced cost importance scaled by objective coefficient
    """
    # Extract the 10 essential features
    obj_coef = variable_features[:, 0]          # Objective coefficient
    reduced_cost = variable_features[:, 7]      # Reduced cost
    fractional = variable_features[:, 9]        # Fractional part of LP solution
    age = variable_features[:, 12]              # Scaled age
    pseudocost_ratio = variable_features[:, 41] # Pseudocost ratio (up/down)
    pseudocost_sum = variable_features[:, 42]   # Pseudocost sum
    constraint_std = variable_features[:, 90]   # Active constraint std (dual_cost weighting)

    # 1. Core fractionality: quadratic function peaking at 0.5
    fractional_score = 4.0 * fractional * (1.0 - fractional)

    # 2. Pseudocost reliability: balanced combination of three pseudocost measures
    pseudocost_score = (params[0] * pseudocost_sum +
                        params[1] * pseudocost_ratio)

    # 3. Constraint tightness indicator (higher std suggests more active constraints)
    constraint_tightness = constraint_std

    # 4. Age penalty: slight preference for recently active variables
    age_factor = 1.0 - 0.3 * age

    # 5. Reduced cost importance scaled by objective coefficient
    reduced_cost_score = np.abs(reduced_cost) * (1.0 + np.abs(obj_coef))

    # Combine components with clear multiplicative structure
    scores = (fractional_score * pseudocost_score * age_factor +
              params[2] * reduced_cost_score +
              0.1 * constraint_tightness)

    return scores
```

