# OpenReview forum: "LLM4Branch: Large Language Model for Discovering Efficient Branching Policies of Integer Programs"
_ICML.cc/2026/Conference — ICML 2026 regular_

### Official Review · Reviewer_aFYB · 2026-02-25

**Soundness:** 4
**Presentation:** 4
**Significance:** 3
**Originality:** 3
**Overall Recommendation:** 5
**Confidence:** 3

**Summary:**

This paper introduces LLM4Branch, a novel framework that automates the discovery of efficient branching policies for Mixed Integer Linear Programming (MILP) solvers.  A critical limitation in current machine learning approaches for branch and bound algorithm is "objective mismatch," where models optimize for surrogate metrics (like accuracy) rather than the solver's end-to-end performance.

To overcome this, LLM4Branch represents a branching policy as an executable Python program consisting of an LLM-generated skeleton and a parameter vector. The framework uses an evolutionary loop where an LLM proposes program skeletons, and a zeroth-order optimization method tunes the parameters using direct performance feedback from the solver on a small set of training instances. Evaluated on the SCIP solver across multiple standard and challenging benchmarks, the discovered CPU-based policies establish a new state-of-the-art over other CPU methods and perform competitively with GPU-heavy models

**Compliance With Llm Reviewing Policy:**

Affirmed.

**Final Justification:**

I think the authors have fully addressed my comment and I will keep my current positive assessment to this paper.

**Key Questions For Authors:**

- How do you explain that LLM4Branch sometimes results in a higher number of nodes compared to other baseline methods? Could this lead to high space complexity (memory usage) for the solver, and what are some potential solutions to address this issue?

- Can you clarify how the complexity of the problems (easy, medium, hard) is defined in terms of parameter scale, and how much this specific scaling impacts the overall branch-and-bound search process?

- While you rely on a single program backbone during testing, would it be feasible to train a portfolio of different programs and dynamically route new MILP instances to the most suitable program?

**Limitations:**

Yes

**Strengths And Weaknesses:**

**Strength**

- The method is highly novel. It successfully uses symbolic program optimization to enhance branch-and-bound heuristics, uniquely combining LLM code generation with parameter tuning .
- The framework shows strong performance and excellent generalization ability using a very small amount of training data, specifically requiring just 8 instances for the evolutionary search .
- This paper is well-structured and presented.

**Weakness**
- The zeroth-order optimization method might have high computational complexity during the training phase. It strictly limits the number of optimizable parameters, which means the approach might struggle to scale to more complex mathematical policies.
- Some very similar concurrent works that also use LLMs to design solver components, such as LLM4Solver and DHEvo, are mentioned in the related work but are not benchmarked or compared against in the main experiments .
- It is not clear that whether these discovered symbolic heuristics generalize well to other proprietary or open-source solver architectures, such as Gurobi or CPLEX, since all evaluations are strictly conducted on SCIP.

---

> ### Author Rebuttal · Authors · 2026-03-31
>
> Thanks for your review. In the following, we address your concerns and questions point by point.
>
> Weaknesses:
>
> A1: We would like to clarify that we intentionally restrict the parameter number
> as the resultant branching policy are to be executed hundreds of thousands of times in the branch-and-bound process of hard instances. From this view, it is not recommended to deploy complex mathematical policies.
>
> A2: We have included a comparison with LLM4Solver on the Setcover dataset. As shown in the table provided in our response to Reviewer FMu5, the empirical demonstrate that LLM4Branch outperforms this baseline across key metrics.
>
>
> A3: We agree with the reviewer that testing our discovered policies on solvers like CPLEX or Gurobi is highly valuable. However, a significant technical hurdle remains: our branching policies require extracting candidate variable features (e.g., our 91-dimensional feature set), which must be developed and integrated within the solver's source code. Unfortunately, Gurobi and CPLEX are closed-source and lack the necessary internal interfaces to extract these features. Regarding other open-source solvers (e.g., HiGHS or CBC), while they are accessible, they often have different internal data structures and interfaces. We plan to further investigate the generalization and testing of our policies on these solver as our future works.
>
> Questions:
>
> A1: Our objective is total solving time rather than purely node reduction. Other baselines, such as GNNs, possess a significantly larger number of parameters and more complex structures, which sometimes allows them to achieve fewer nodes. However, the advantage of our method lies in its lower computational overhead. Even when the node count is slightly higher, the decision-making per node is much faster, leading to a shorter total solving time.
>
> To address the concern of high space complexity, we designed a variant called **LLM4Branch-Hybrid**. Specifically, at lower depths (depth $\le$ 2), we employ strong branching policy; In the rest, we switch to our discovered policy. As shown in the table below, this hybrid method effectively reduces node number while it incurs higher solving time. In the final version, we will include the LLM4Branch-Hybrid variant and discuss other potential solutions to address memory issues.
>
> Table link: https://anonymous.4open.science/r/anonymous20992134/hybrid.pdf
>
> A2: As detailed in Appendix C (Table 9), we follow the complexity definitions in [1, 2, 3] for a fair comparison. As these parameters scale, the branch-and-bound process faces challenges, such as increased candidate branching variables and increased per-node LP solving time. This makes the design of an effective branching policy much more challenging: the larger pool of candidate variables complicates decision-making, and the high per-node cost penalizes the suboptimal branching choices that unnecessarily expand the search tree. Importantly, our discovered policies are independent of the problem size. They are trained on Easy instances and directly applied to unseen Medium and Hard instances.
>
> A3: This is an excellent suggestion. We conducted a preliminary analysis by randomly selecting eight instances from the Setcover Medium dataset and evaluating the performance of different programs generated by LLM4Branch. As shown in the table below, different programs exhibit varying strengths across individual instances. For example, Program 1 excels on Instances 1 and 6, while Program 2 achieves the best performance on Instance 4.
>
> Table link: https://anonymous.4open.science/r/anonymous20992134/router.pdf
>
> This performance gap suggests that a portfolio-based approach which routes instances to the most suitable program using a lightweight classifier could further improve results. We will add a detailed discussion of this potential in Appendix G and highlighted it as a key direction for future work.
>
> References:
>
> [1] Gupta et al., Hybrid models for learning to branch. NeurIPS, 2020.
>
> [2] Feng et al., Sorrel: Suboptimal-demonstration guided reinforcement learning for learning to branch. AAAI, 2025.
>
> [3] Kuang et al., Rethinking branching on exact combinatorial optimization solver: The first deep symbolic discovery framework. ICLR, 2024.

---

> > ### Author Rebuttal · Reviewer_aFYB · 2026-04-01
> >
> > Thank you for the detailed rebuttal and for thoughtfully addressing my questions regarding space complexity, problem scaling, and portfolio-based routing. My overall assessment of the paper is unchanged, and I will keep my score of 5 for acceptance.

---

> > > ### Author Response · Authors · 2026-04-08
> > >
> > > Thank you for reviewing our work. We are pleased our responses addressed your questions and appreciate your support.

---

### Official Review · Reviewer_8nv3 · 2026-03-05

**Soundness:** 2
**Presentation:** 2
**Significance:** 1
**Originality:** 3
**Overall Recommendation:** 2
**Confidence:** 4

**Summary:**

This paper presents LLM4Branch, a novel framework that leverages Large Language Models to automate the discovery of efficient branching policies for Mixed Integer Linear Programming (MILP) solvers. The approach represents branching policies as executable programs with LLM-generated skeletons and zeroth-order optimized parameters.

**Compliance With Llm Reviewing Policy:**

Affirmed.

**Final Justification:**

I will keep my score and will not consider the proposed method novel enough to be accepted.

**Key Questions For Authors:**

1. What LLM for coding did you use in this paper?

**Limitations:**

1. Using vibe-coding to solve combinatorial problems is great, but the scientific paper needs to provide more than that.

**Strengths And Weaknesses:**

# Strength
1. Leverage LLM to generate branching skeleton programs or code directly is novel. It successfully adapts the LLM-based evolutionary discovery paradigm to the specific, long-standing challenge of automating branching policy design.

# Weakness
1. What is the scientific discovery this paper provided? The vibe-coding capabilities of LLM are great, and the branching policy it discovered is good. But the reason why it works needs to be explored.
2. The method is trained on a very small set of instances (e.g., 8). While it shows good generalization, optimizing solely for performance on these few "Easy" instances could lead to policies that exploit instance-specific shortcuts, which may not scale optimally.

---

> ### Author Rebuttal · Authors · 2026-03-31
>
> Thanks for your review. In the following, we address your concerns and questions point by point.
>
> Strengths And Weaknesses:
>
> A1: We thank the reviewer for this perspective on the scientific depth of our work. Regretably, we are unable to provide a formal theoretical proof for the reason why it works. Kindly remind that the interpretability of LLMs remains a well-known open challenge in the community. Unlike "vibe-coding", our framework incorporates specific MILP domain knowledge and solver feedback to guide the LLM to refine its generated policy. This makes the discovery process a efficient exploration of the policy space rather than a stochastic "vibe-based" attempt. Moreover, as discussed in Appendix G, the discovered policies are human-readable and providing new insights into how specific feature couplings can help enhance efficiency of branch-and-bound process. These findings also offer new inspirations for the future design of branching heuristics.
>
> A2: We agree that training on few instances may not guarantee optimality. Instead, our objective is to rapidly generating high-quality heuristics at an acceptable cost. The use of 8 instances is a good design choice that aligns with this goal, made possible by our compact symbolic representation. Unlike neural networks with millions of parameters, our policies typically use fewer than 10 parameters. This acts as a powerful regularizer against overfitting. Our empirical results in Table 10 of Appendix G support this, demonstrating strong cross-benchmark generalization (e.g., a policy evolved on SetCover performs excellently on Cauctions). This confirms that LLM4Branch discovers fundamental branching principles rather than exploiting instance-specific shortcuts.
>
> Key Questions For Authors:
>
> A1: As noted in Section 5.1 (Page 7, Left Column, Line 360), we used **DeepSeek-R1** to generate branching policies. Furthermore, to ensure the robustness of our framework, we also conducted comparative experiments across various LLMs. The results are detailed in Table 6, which demonstrates the consistency of our evolutionary framework across different backends.
>
> Limitation:
>
> A1: We clarify that our method is not "vibe-coding" through one-shot generation. Instead, we designed a evolutionary framework that leverages LLMs to automate the discovery of efficient branching policies. The LLM iteratively refines the branching policy based on solver feedback. This evolutionary process ensures that the discovered policies are not "lucky guesses" or mere "vibes", but are **verified** and **efficient**.

---

> > ### Author Rebuttal · Reviewer_8nv3 · 2026-04-03
> >
> > Since agentic coding generation and refinement are go-to methods for algorithms like Claude Code. I will not consider the proposed method novel enough to be accepted. I will keep my score.

---

> > > ### Author Response · Authors · 2026-04-08
> > >
> > > We sincerely thank the reviewer for the continued engagement and for sharing this perspective. We understand how, at a high level, using an LLM to generate executable functions might seem conceptually similar to general coding agents like Claude Code. However, we would like to respectfully clarify the unique technical challenges in exact combinatorial optimization that necessitate our specific design, distinguishing LLM4Branch from general software engineering tools.
> > >
> > > We hope to highlight why integrating LLMs into MILP solvers represents a methodological contribution to the ICML community:
> > >
> > > - Domain-Specific Optimization: Discovering an efficient MILP branching policy is a highly complex mathematical problem. The feedback from a Branch-and-Bound solver is essentially a black box with delayed rewards (e.g., node counts or solving time). A general coding agent cannot natively evaluate or optimize continuous algorithmic parameters against such delayed, non-differentiable metrics.
> > >
> > > - The Novel Two-Phase Framework: The core innovation of LLM4Branch is not merely "prompting an LLM to write code," but the explicitly designed Two-Phase framework. We purposefully decouple the massive search space: the LLM is restricted to the program search (Program Skeleton Generation), while a Zeroth-Order Optimization method tunes the continuous parameters. This specific framework is the key that enables us to bypass the objective mismatch problem inherent in previous imitation approaches learning from strong branching.
> > >
> > > - Overcoming the "End-to-End" Bottleneck: Leveraging end-to-end solver feedback to discover branching rules has been severely bottlenecked by sample inefficiency in Reinforcement Learning. Our framework innovatively reframes this challenge into an evolutionary program search. By doing so, we achieve State-of-the-Art (SOTA) performance among CPU-based policies using only 8 training instances—an empirical breakthrough for practical, hardware-efficient solver design.
> > >
> > > We draw deep inspiration from recent pioneering works like FunSearch (Nature 2024) and AlphaEvolve (2025), which demonstrate that adapting LLMs for algorithmic and heuristic discovery is a distinct and highly impactful research frontier. LLM4Branch aims to advance this frontier by penetrating the core components of exact mathematical solvers.
> > >
> > > We genuinely appreciate the reviewer's feedback, and we hope this clarification helps the Area Chair and all reviewers further appreciate the specialized framework design and its empirical value that brings to the combinatorial optimization community.

---

### Official Review · Reviewer_m6HM · 2026-03-11

**Soundness:** 3
**Presentation:** 4
**Significance:** 3
**Originality:** 3
**Overall Recommendation:** 4
**Confidence:** 3

**Summary:**

This paper addresses the limitations of current learning-based Mixed Integer Linear Programming (MILP) solvers, specifically their heavy reliance on costly expert demonstrations and the gap between training objectives and end-to-end performance. The authors propose a framework that leverages Large Language Models to generate algorithmic structures, which are then refined through a Historical Policy-Performance Database and a multi-step filtering process. Finally, the generated programs are optimized via parameter tuning.

**Compliance With Llm Reviewing Policy:**

Affirmed.

**Final Justification:**

Thank you to the authors for their thoughtful revisions and clarifications, which have improved the overall quality and presentation of the paper. The work addresses an interesting problem and demonstrates solid technical effort. However, I still have some remaining concerns, and therefore maintain my original score.

**Key Questions For Authors:**

1. The current iterative optimization process appears to focus on refining code from the previous iteration. Does this localized search strategy risk converging to sub-optimal local minima? Does the method suffer from performance degradation?
2. The framework relies on LLMs for multi-step code generation, which inherently involves significant token consumption. Does this then imply that the method proposed in the paper is actually significantly more costly than the baseline?
3. The authors should distinguish the framework's contribution from the LLM's inherent capabilities. Does the model size have a significant impact on this method?

**Limitations:**

Yes

**Strengths And Weaknesses:**

Strengths：
1. The authors provide sufficient implementation details, allowing readers to intuitively understand the underlying logic and operational workflow of the proposed system.
2. The authors demonstrate the method's effectiveness across extensive benchmarks. Their approach generalizes well to unseen instances. Additionally, the paper transparently discusses current limitations. It highlights the performance gap compared to GNN-based GPU methods.

Weaknesses：
1. The method‘s success depends on the power of the underlying LLM. The framework relies heavily on the model's inherent capabilities.
2. The proposed process uses multi-step filtering. It also uses iterative optimization. These steps may cause significant computational overhead. The search phase could also lead to high costs.

---

> ### Author Rebuttal · Authors · 2026-03-31
>
> Thanks for your review. In the following, we address your concerns and questions point by point.
>
> ## LLM Dependence and Impact of Model Size (Weakness 1 & Question 3)
> As demonstrated in Table 6, our framework maintains consistently high performance whether utilizing DeepSeek-R1, GPT-5, DeepSeek-V3, or Qwen3-Next-80B-A3B. This showns the robustness of our evolutionary loop across these models. To further investigate whether our method heavily relies on massive model size, we extended our evaluation to include a smaller model, Qwen3-32B. As shown in the table below (**LLM4Branch-Qwen3**), LLM4Branch powered by Qwen3-32B still achieves highly competitive performance, consistently outperforming the solver's default RPB method.
>
> To distinguish the framework's contribution from the LLM's inherent capabilities, we established a baseline only using LLM. In this setup, we prompted both DeepSeek-R1 and Qwen3-32B to directly generate 200 independent branching policies. To ensure a fair comparison, these models were provided with the system prompt defined in Appendix D.1, including task instructions, feature descriptions, and formatting requirements. We then evaluated all 200 generated policies on the 8 training instances and selected the single best-performing policy based on solving time. For notational simplicity, we denote this baseline as Direct Sampling (DS). As shown in the table below, our framework significantly outperforms the DS approach. This comparison shows that the performance gains stem from our evolutionary refinement and rigorous parameter optimization.
>
> | Setcover | Easy | | Medium | | Hard | |
> | :--- | :--- | :--- | :--- | :--- | :--- | :--- |
> | **Model** | **Time** | **Nodes** | **Time** | **Nodes** | **Time** | **Nodes** |
> | RPB | 5.67 | 55.16 | 53.83 | 2474.10 | 982.84 | 75384.37 |
> | LLM4Branch-R1 | 3.21 | 166.49 | 37.64 | 2724.87 | 952.92 | 64479.94 |
> | LLM4Branch-Qwen3 | 3.31 | 171.83 | 39.14 | 2980.07 | 974.81 | 67328.38 |
> | LLM4Branch-R1-DS | 3.58 | 206.82 | 47.64 | 3869.02 | 1054.91 | 79247.43 |
> | LLM4Branch-Qwen3-DS | 3.67 | 220.35 | 57.06 | 4993.89 | 1187.22 | 81312.31 |
>
> ## Token Consumption and Computational Overhead (Weakness 2 & Question 2)
>
> To address your concerns, we provide a comprehensive breakdown of the costs as follows:
>
> **Token Cost** As explicitly documented in our paper (Appendix E, under "Running Cost"), the token consumption is economical. A complete evolutionary search run consumes approximately 1.2 million tokens, which costs only about $0.20 using DeepSeek-R1.
>
> **Training Time** Our method’s runtime is well within a typical range of, and entirely comparable to, other learning-based baselines. For instance, on the SetCover benchmark, a complete run of LLM4Branch takes approximately 6.0 hours while the training times for the baselines on the same hardware are: Symb4CO (1.2h), MLP (2.5h), Hybrid (8.7h), and GNN (10.1h).
>
> **Cost of Expert Labeling** Kindly note that comparing only model training time may underestimate the true computational cost of the baselines. Baselines (like MLP, GNN, and Hybrid) require an expensive data labeling phase to extract strong branching targets. As highlighted in recent literature [1], this labeling scheme scales exponentially. For example, collecting 120,000 samples for larger SetCover instances can take dozens of days of CPU time. By contrast, LLM4Branch completely bypasses this process.
>
> ## Risk of Local Minima (Question 1)
>
> As training neural networks, finding the global optimum within the vast space of branching programs is computationally intractable. Therefore, we acknowledge the risk of converging to sub-optimal local minima. To reduce this risk, we adopt a commonly used exploration strategy [2]. Specifically, as detailed in Appendix D.2, our framework periodically samples from the entire historical database, rather than solely refining the previous iteration. This mechanism continuously improves structural diversity, helping the search escape local minima.
>
> Empirically, we observe that our framework progressively discovers better programs without performance degradation. As shown in Figure 2, within the given computational budget of 200 iterations, the evolutionary search demonstrates a consistent trend of performance improvement.
>
>
> Reference:
>
> [1] Parsonson et al., Reinforcement learning for branch-and-bound optimisation using retrospective trajectories, AAAI 2023.
>
> [2] Shojaee et al., LLM-SR: Scientific equation discovery via programming with large language models, ICLR 2025.

---

> > ### Author Rebuttal · Reviewer_m6HM · 2026-04-04
> >
> > Thanks for clarifying. I will maintain my score.

---

> > > ### Author Response · Authors · 2026-04-08
> > >
> > > Thank you for reviewing our work. We are pleased our responses addressed your questions and appreciate your support.

---

### Official Review · Reviewer_FMu5 · 2026-03-13

**Soundness:** 3
**Presentation:** 2
**Significance:** 2
**Originality:** 3
**Overall Recommendation:** 4
**Confidence:** 1

**Summary:**

The paper proposes LLM4Branch, an end-to-end framework that leverages LLMs to discover efficient branching policies for MILP solvers automatically. The authors represent the branching policy as an executable program. Using this representation, an LLM generates the human-readable program skeleton to define the branching policy structure, while a zeroth-order optimizer tunes some of the parameters based on solver feedback.

**Compliance With Llm Reviewing Policy:**

Affirmed.

**Final Justification:**

The additional experiments and clarifications are helpful and help me better understand their work. I've raised my score.

**Key Questions For Authors:**

1. As mentioned in Section 5, the end-to-end optimization objective is designed to minimize the geometric mean number of branching nodes across the training instances, have you considered using other criteria? How would the performance and stability be?

2. All experiments are conducted on setting with only eight per benchmark for the evolutionary search, but results under larger benchmarks are lacking. For example, does the performance increases or does overfitting occur when the number of benchmarks increases from 8 to 16 or even larger? Are there a "minimum necessary size"?

**Limitations:**

See “Weaknesses”.

**Strengths And Weaknesses:**

Strengths:

1. The proposed method is purely CPU-based but achieves competitive performance with GPU-based models.

2. The method demonstrates good performance with very little training data.

Weaknesses:

1. Some parts of the writing could be improved. For example, the function code inserted in Section 4.1 is difficult to read.

2. The proposed method could be further investigated, for example, current optimization objective is to minimize the number of nodes, but this metric is not entirely equivalent to the solving time. Some branching policies may require fewer nodes but have high per-node costs. Further investigation on different metrics co-operation or adaptive metric switching would be beneficial.

3. The selected baselines are limited: only one baseline was proposed in 2024, while the others are from 2019 and 2020. Are there any related works proposed between 2020 and 2024, or even more recent ones?

---

> ### Author Rebuttal · Authors · 2026-03-31
>
> Thanks for your review. In the following, we address your concerns and questions point by point.
>
> Weaknesses:
>
> A1: We will improve the writing of the paper. Moreover, we will reformat the code block in Section 4.1 and add more explanatory comments to improve readability.
>
> A2: We fully agree that the number of branching nodes is not entirely equivalent to the actual solving time. As we briefly discussed in Appendix G (Page 18, Lines 972-978), minimizing the number of nodes allows for a purely branching policy refinement. While solving time reflects the quality of the branching policy, it is also heavily hardware-dependent, which introduces stochastic noise that may mislead the LLM's evolutionary search with "false improvements". Consequently, as in other works [1, 2], we use node count as a hardware-independent metric for optimization.  We will clarify this point in the revised Appendix G.
>
> Indeed, we agree that  exploring metric co-operation or adaptive metric switching is beneficial which definitely deserves further investigation in the revison. Thanks a lot!
>
> A3: Following your valuable suggestion, we have conducted additional comparative experiments against two recent baselines that we discussed in our Related Work section: LLM4Solver (2024) and SORREL (2025) [2, 3]. Given the time limit of rebuttal, we can only report a comparative evaluation on the Setcover benchmark and other benchmarks will be included in the final revision. The results below show that LLM4Branch consistently outperforms both methods. Specifically, LLM4Branch reduces the solving time by 11.1% ~ 39.9% compared to LLM4Solver, and by 9.1% ~ 34.5% compared to SORREL.
>
> | Setcover | Easy | | Medium | | Hard | |
> | :--- | :--- | :--- | :--- | :--- | :--- | :--- |
> | **Model** | **Time** | **Nodes** | **Time** | **Nodes** | **Time** | **Nodes** |
> | SORREL | 3.53 | 174.30 | 57.47 | 3575.97 | 1172.81 | 67181.72 |
> | LLM4Solver | 3.61 | 212.68 | 62.59 | 3833.70 | 1348.09 | 73484.81 |
> | LLM4Branch | **3.21** | **166.49** | **37.64** | **2724.87** | **952.92** | **64479.94** |
>
> Questions:
>
> A1: To address your question, we compared the Geometric Mean (GM) against the Arithmetic Mean (AM), the worst-case performance (Max), and the best-case performance (Min) across the 8 training instances. We ran each configuration 5 times with different random seeds to report the mean and standard deviation. As shown in the empirical results, both GM and AM achieve the best performance and highest stability. In contrast, optimizing for Max or Min leads to noticeable performance degradation and higher variance. One reason is that the Max criterion forces the evolutionary search to optimize for the single hardest training instance, while the Min criterion solely focuses on the easiest instance, thereby failing to discover robust and generalized branching policies. Therefore, we maintain GM as our objective and will report this in the revision.
>
> Table link: https://anonymous.4open.science/r/anonymous20992134/criteria.pdf
>
> A2: While training on more instances may improve policy generalization, it also increases the computational cost of our evolutionary search. Strictly Speaking, it is difficult to define a "minimum necessary size". To illustrate this trade-off, we conducted a comparative study using 8 (our default), 16, and 32 instances. As shown in the table below, we found that increasing the number of instances to 16 and 32 only brings marginal improvements. Consequently, we selected 8 instances.
>
> Table link: https://anonymous.4open.science/r/anonymous20992134/instances.pdf
>
> References:
>
> [1] Parsonson et al., Reinforcement learning for branch-and-bound optimisation using retrospective trajectories, AAAI 2023.
>
> [2] Feng et al., Sorrel: Suboptimal-demonstration guided reinforcement learning for learning to branch. AAAI, 2025.
>
> [3] Zhou et al., LLM4Solver: Large language model for efficient algorithm design of combinatorial optimization solver. 2024. URL https://openreview.net/pdf?id=XTxdDEFR6D.

---

> > ### Author Rebuttal · Reviewer_FMu5 · 2026-04-03
> >
> > Thank you for the additional experiments and clarifications. They are helpful and I'm willing to raise my score.

---

> > > ### Author Response · Authors · 2026-04-03
> > >
> > > Thanks for your positive feedback. We are glad that our additional experiments and clarifications successfully addressed your concerns.
> > >
> > > We noticed that the score remains unchanged on the system. We understand you might be busy, but we would kindly appreciate it if you could update the score at your earliest convenience to reflect your current evaluation.

---

### Decision · Program_Chairs · 2026-04-30

**Decision:**

Accept (regular)

**Comment:**

Three reviewers support this work with low confidence, and another reviewer rejects this work with high confidence. I read through the paper and rebuttal, I found this paper is interesting with some commendable designs, which were different from the existing LLM4MIP works, and may offer some insight for subsequent researchers of this topic. I also respect the decision from the negative reviewer, however, it seems his/her comments are relatively skinny and intangible. While I recommend 'weak accept', the authors please, 1) include more LLM baselines and results, 2) explicitly discuss the comment from the negative reviewer in the final version.